# LARP: Learner-Agnostic Robust Data Prefiltering

**Kristian Minchev**                                    *kristian.minchev@insait.ai*
*INSAIT, Sofia University "St. Kliment Ohridski"*

**Dimitar I. Dimitrov**                                *dimitar.iliev.dimitrov@insait.ai*
*INSAIT, Sofia University "St. Kliment Ohridski"*

**Nikola Konstantinov**                              *nikola.konstantinov@insait.ai*
*INSAIT, Sofia University "St. Kliment Ohridski"*

**Reviewed on OpenReview:** *https://openreview.net/forum?id=gI6VOV3jfO*

## Abstract

Public datasets, crucial for modern machine learning and statistical inference, often contain low-quality or contaminated samples that can harm model performance. This creates a need for principled prefiltering procedures that a data provider can apply to protect the accuracy of a range of potential downstream statistical and learning procedures *simultaneously*. In this work, we formalize and analyze **L**earner-**A**gnostic **R**obust data **P**refiltering (LARP), the problem of designing prefiltering procedures with guarantees on the worst-case loss over a pre-specified set of learners. We establish the feasibility of LARP in two theoretical settings, by providing upper-bound guarantees on the worst-case loss. Our theoretical results indicate that protecting heterogeneous learner sets via LARP comes at the price of some performance loss compared to individual, learner-specific prefiltering; we call this gap the price of LARP. To assess this gap in performance, we empirically measure the price of LARP across image and tabular tasks. We further explore potential benefits of LARP from the perspective of saving on repeated data curation efforts, in a game-theoretic model where the downstream learners can split the cost of the single prefiltering.

## 1 Introduction

The availability of large, public datasets has underpinned recent successes of statistical and machine learning methods. For example, public benchmarks such as ImageNet (Deng et al., 2009) and GLUE (Wang et al., 2019) have enabled the creation of numerous pre-trained and competitive models, readily available for adoption and fine-tuning by practitioners in any industry. Similarly, public healthcare datasets, e.g., the public data on the COVID-19 pandemic released by the WHO[1], can serve as a valuable reference to medical professionals for estimating statistics about common diseases and treatments.

Despite their attractiveness, public datasets are often susceptible to noisy, inaccurate or even maliciously manipulated data (Carlini et al., 2024). Since statistical methods are vulnerable to data contamination, such issues can damage the accuracy of downstream learning procedures applied on top of public data. For example, large web corpora used for language-model pretraining have been found to contain some toxic, biased, and otherwise problematic samples (Gehman et al., 2020; Dodge et al., 2021; Luccioni & Viviano, 2021). Models trained on such data have consequently been shown to sometimes produce toxic or biased outputs when prompted (Gehman et al., 2020). Similarly, the LAION-5B dataset (Schuhmann et al., 2022), a widely-used open image-text corpus, has been found to contain some synthetic images that can harm downstream model training (Alemohammad et al., 2023). Another impact of public data inaccuracies manifested during the COVID-19 pandemic. Early public data often had to be collected based on limited testing capacity and early estimates of the disease mortality were later reported to be underestimates (Msemburi et al., 2023).

---

[1] https://data.who.int/dashboards/covid19/data

These issues motivate the question of how public datasets can be prefiltered by the data provider, so as to explicitly protect the accuracy of downstream statistical and learning procedures (hereafter referred to as *learners*, for brevity) applied on top of the dataset. Such prefiltering poses a new technical challenge for the data provider: designing data prefiltering algorithms that are *learner-agnostic*, in the sense of protecting the accuracy of a wide specification of learners simultaneously. While learner-agnostic prefiltering naturally constitutes a harder statistical challenge than prefiltering to protect a single learning algorithm, it is intuitively desirable and also aligns with recent calls for transparency in dataset creation (Gebru et al., 2021) and for data-centric strategies in developing robust and ethical AI systems (Liang et al., 2022). Last but not least, learner-agnostic prefiltering can benefit downstream dataset users through reduced data curation efforts, which can be substantial and expertise-demanding at the scale of state-of-the-art ML benchmarks and datasets.

**Contributions** In this work we study the problem of prefiltering public data to protect the accuracy of a wide specification of learners, proposing a framework of *Learner-Agnostic Robust data Prefiltering (LARP)*. We formalize this as the task of finding a prefiltering procedure with guarantees on a learner-agnostic risk, defined as the worst-case loss over a pre-specified set of learners.

We first study the feasibility of the LARP objective. In the context of scalar mean estimation, we provide a problem instance which highlights that non-trivial bounds on the learner-agnostic risk necessarily depend on the learner set. Then we prove an upper bound on the learner-agnostic risk over any set of Huber estimators (Huber, 1964), achieved by a prefiltering procedure based on popular methods for outlier removal using quantiles. The bound contains two terms: one reminiscent of standard results in robust statistics, featuring the data corruption rate and the dataset size; and one that features the dependence on the learner set. We also present a PAC-style setup in which we use abstract downstream learners that are characterized only by an upper bound on the true risk of their respective hypotheses. We use an idealized prefiltering procedure in order to provide upper bounds which depend on the effectiveness of the prefiltering.

Our theoretical results suggest an inherent trade-off when performing LARP: protecting a broader set of learners via the single prefiltering leads to more pessimistic accuracy guarantees compared to performing individual prefiltering for each learner. We call the average utility reduction across the learner set the *price of LARP* and study it via extensive experiments on the Adult (Becker & Kohavi, 1996) and CIFAR-10 (Krizhevsky, 2009) datasets with label noise. We make our code publicly available[2]. We also provide a modification of our framework that captures varying risk functions in downstream learners such as fairness and accuracy. Finally, we compare the price of LARP to potential benefits from the perspective of saving on repeated data curation efforts, by studying a game-theoretic setting where the downstream learners can split the cost of the single prefiltering.

## 2 Related work

We are, to our knowledge, the first to give a theoretical framework for data prefiltering to protect the downstream accuracy of a specified learner set. To put our work in perspective, we survey relevant works and discuss similarities and differences with our approach.

**Robust statistics and learning** Learning from contaminated data is a classic problem in robust statistics and machine learning (Huber, 2004; Cinà et al., 2023). Numerous works have studied various contamination models in the context of statistical estimation (Huber, 1964; 2004; Kearns & Li, 1988; Diakonikolas et al., 2019a; Kane et al., 2024; Diakonikolas & Kane, 2023) and supervised learning (Kearns & Li, 1988; Biggio et al., 2012; Diakonikolas et al., 2019b). Some commonly studied types of data corruption are label noise (Natarajan et al., 2013; Patrini et al., 2017; Han et al., 2018; Northcutt et al., 2021b; Zhang & Sabuncu, 2018) and shortcuts (Geirhos et al., 2020; Shah et al., 2020; Nam et al., 2020; Sagawa et al., 2020b) which we consider in our experiments.

---

[2]https://github.com/insait-institute/LARP

Unlike robust statistics/ML, however, in which one finds robust learners, we focus on finding robust data prefiltering procedures for a fixed learner set. As we show in Section 3, this leads to a new optimization problem where the solution depends on the pre-specified set of learning algorithms.

**Learning robustly under multiple distributions**  Our work is concerned with a minimax bound over a set of downstream risks, and thus is conceptually similar to distributionally-robust optimization (DRO) (Ben-Tal et al., 2013; Sagawa et al., 2020a) and multi-distribution learning (MDL) (Blum et al., 2017; Haghtalab et al., 2022). Such frameworks optimize a single learner to ensure robustness of multiple data distributions (for MDL) or perturbed test sets (for DRO). In LARP, on the other hand, one optimizes over what data to keep (through the prefiltering mechanism) to ensure robustness of multiple downstream learners.

Just as the DRO literature considers the "price of robustness" (Bertsimas & Sim, 2004), quantifying the reduction in a model's standard accuracy due to increased distributional robustness, LARP incurs a "price of LARP" (see Section 3) due to the prefiltering being protective for a larger variety of learners. Apart from the benefits of providing guarantees for multiple learners, we argue that LARP can be beneficial for the sake of saving on total data curation costs. Specifically, in Section 5.3 we study within a game-theoretic model whether cost savings from splitting the prefiltering cost between downstream dataset users can offset utility reduction stemming from the price of LARP.

**Data-centric ML**  Data-centric machine learning focuses on the critical role of data quality for model performance. Surveys by Zha et al. (2025) and Whang et al. (2023) provide comprehensive overviews of methodologies aimed at improving data quality through validation, cleaning, and maintenance. One line of work studies learner-agnostic methods in the context of data valuation (Just et al., 2023; Kessler et al., 2025). Another approach, which is focused on mitigating label noise, is the data pruning strategy (Park et al., 2023). Our work introduces a new objective for this task: minimizing the maximum loss across a pre-specified set of learners.

**Data moderation**  A large body of literature studies methods for data quality control and moderation for ML data obtained via crowdsourcing (Lease, 2011; Awasthi et al., 2017; Vaughan, 2018; Sheng & Zhang, 2019). These works study specific models of data contamination, e.g., label noise, and develop techniques for data filtering with provable guarantees on the resulting data quality. Unlike these works, we only adopt limited assumptions on the type of corruption and seek to certify the quality of downstream learned models directly.

Another line of work studies the impact of data moderation in the context of human learning and disinformation (Haghtalab et al., 2021; Dwork et al., 2024; Huleihel & Refael, 2024). The social implications in such settings lead to many orthogonal considerations for moderators, in particular designing appropriate models of human learning, reducing polarization and diversity of the provided information. In contrast, we focus on protecting the accuracy of downstream learning algorithms in the presence of corrupted data.

**Fragmented learning pipelines**  Our work is motivated by the increasing availability of open-access datasets for training, which necessitates their preprocessing and preparation for public use. This is an example of the increasing *fragmentation* of modern learning pipelines, as multiple stakeholders become involved in one or more stages, including data gathering, preprocessing, training and deployment. Several works focus on the interactions between data providers and model creators in ML. Data delegation and valuation have been studied by Chen et al. (2022); Saig et al. (2023); Ananthakrishnan et al. (2024), with the aim of developing appropriate contract theory and fair remuneration methods for outsourcing data-related tasks. Other works consider strategic interactions between foundation model creators and entities performing fine-tuning (Laufer et al., 2024). Our work addresses a concrete form of pipeline fragmentation where data preprocessing is performed independently of model training. We adopt the perspective of a data moderator tasked with ensuring downstream robustness.

## 3 Framework

In this section we formalize the problem of finding learner-agnostic robust prefiltering procedures. First, we define the environment of the problem, which consists of the contamination model and the set of downstream learners. Then we move on to present the main objective of LARP. Finally, we discuss differences with classic robust learning and we motivate and define a notion of price of LARP.

**Learning setup**  We consider the general data contamination model from robust learning. In the canonical setting of learning theory, one is interested in learning a property $\theta \in \Theta$ of a distribution $\mathcal{D}_\theta$. However, instead of i.i.d. data from $\mathcal{D}_\theta$, in the robust setting one assumes that an arbitrary $\epsilon$-fraction of the points is corrupted according to some contamination model. We refer to $\epsilon \in [0, 1]$ as the contamination rate. We note that many contamination models require that $\epsilon < 1/2$ in order to provide any meaningful learning guarantees (Diakonikolas & Kane, 2023).

Each downstream learner $l : \cup_{n=1}^\infty \mathcal{X}^n \to \mathcal{H}$ maps a finite sample $S$ to a hypothesis $\hat{h} = l(S)$. For example, we have $\mathcal{H} = \mathbb{R}$ for scalar mean estimation and $\mathcal{H} = \{h : \mathcal{X} \to \{0, \ldots, c-1\}\}$ for classification. We denote the set of learners as $\mathcal{L}$.

**Learner-agnostic robust prefiltering**  We model a situation where a data provider seeks to prefilter a dataset prior to public release, so that multiple downstream learning algorithms enjoy performance guarantees. Specifically, consider an $\epsilon$-contaminated dataset $S$. Given this dataset, a prefiltering procedure is a function $F : \cup_{n=1}^\infty \mathcal{X}^n \to \cup_{n=1}^\infty \mathcal{X}^n$ satisfying $F(S) \subseteq S$. The prefiltered subset $S' = F(S)$ is then presented to the set of downstream learners, where each $l \in \mathcal{L}$ produces a hypothesis $l(S')$. The performance of each hypothesis is then measured using a risk function $R(l(S'), \theta)$.

Our goal is to design prefiltering procedures that protect all learners in $\mathcal{L}$. Formally, we aim to minimize the *learner-agnostic risk*

$$R_{agn}(F) := \max_{l \in \mathcal{L}} R(l(F(S)), \theta). \tag{1}$$

over a set of possible prefiltering procedures $\mathcal{F}$.

For brevity, we denote $R_l(F) := R(l(F(S)), \theta)$. Classic examples of risk functions include the squared loss $R(\hat{h}, \theta) = (\hat{h} - \theta)^2$ for scalar mean estimation, or population risk $R(\hat{h}, \theta) = \mathbb{P}_{(X,Y) \sim \mathcal{D}_\theta}(\hat{h}(X) \neq Y)$ for classification tasks. Although we assign the same risk function to all learners, in Section 5.2 we also consider a modification of our framework where downstream learners differ in their risk functions instead of their learning algorithms. Finally, note that $R_{agn}$ is a random variable as it depends on the sample $S$. In this work, we focus on high-probability bounds for $R_{agn}$.

**Analyzing the learner-agnostic risk**  Although our setup is concerned with producing accurate hypotheses from corrupted data, similarly to classical robust learning, there are important conceptual differences. In robust learning one aims to find $\arg\min_l R(l(S), \theta)$ over all possible learners $l : \cup_{n=1}^\infty \mathcal{X}^n \to \mathcal{H}$. The main difference with our framework is that in the LARP problem the learner set $\mathcal{L}$ is fixed, and instead minimization happens over the prefiltering procedure $F$. Thus, the optimal procedure may vary significantly according to the properties of the downstream learners. This can be highlighted by rewriting Eq. (1) as

$$R_{agn}(F) = \min_{l \in \mathcal{L}} R_l(F) + \left( \max_{l \in \mathcal{L}} R_l(F) - \min_{l \in \mathcal{L}} R_l(F) \right). \tag{2}$$

The first term, $\min_{l \in \mathcal{L}} R_l(F)$, is the best error that can be achieved by any learner in $\mathcal{L}$. Under the assumption that at least one learner in $\mathcal{L}$ is reasonable, this resembles the target objective of classic robust learning. The second term encapsulates the hardness that arises from the heterogeneity between the learners. In the general case, this term is positive and induces additional losses not present in standard robust frameworks. In Section 4 we study the objective $R_{agn}(F)$ in further detail in two theoretical setups.

**Price of learner-agnostic prefiltering**  The second term in Eq. (2) indicates that applying LARP for larger learner sets may lead to worse downstream performance. This is intuitive, as prefiltering a public

dataset with the goal of protecting the performance of *multiple learning procedures* is intuitively harder than prefiltering the dataset *for a specific learner* (i.e., prefiltering performed by anyone using the dataset for their own use-case).

We quantify this utility loss via the "price of learner-agnostic prefiltering". Formally, we call an instance of LARP *learner-specific prefiltering for l* if $\mathcal{L} = \{l\}$. We want to compare the across-learner performance of a prefiltering procedure $F \in \mathcal{F}$ selected by LARP on a learner set $\mathcal{L}$, to the performance of the learner-specific optimal prefiltering procedures $F_l^* = \arg\min_{F \in \mathcal{F}} R_l(F)$ for each $l \in \mathcal{L}$. Note that $R_l(F) \geq R_l(F_l^*), \forall F \in \mathcal{F}, l \in \mathcal{L}$, which leads to each learner losing utility which we describe using a function $\mathcal{U}_{red} : \mathbb{R}^2 \to \mathbb{R}$ [3]. We define *the price of learner-agnostic prefiltering* of a prefiltering procedure $F$ as

$$P(F) := \frac{1}{|\mathcal{L}|} \sum_{l \in \mathcal{L}} \mathcal{U}_{red}(R_l(F), R_l(F_l^*)). \tag{3}$$

Despite the loss in utility, LARP can still be beneficial for two reasons. Firstly, LARP provides guarantees for a wide range of learners, rather than just one, which might be desirable from the perspective of the data provider. Secondly, it is intuitively "cheaper" to prefilter a dataset once, than for each learner individually. Therefore, in Section 5 we measure the price of learner-agnostic prefiltering in realistic setups and study incentives for performing LARP over learner-specific prefiltering, from the perspective of saving on total data curation costs.

# 4 Theoretical analysis of learner-agnostic risk

In this section we analyze the problem of LARP in two theoretical setups. We first consider the task of scalar mean estimation, providing a hardness result and an upper bound for the learner-agnostic risk. Then, we provide a PAC-style setup which generates a family of LARP instances. We provide guarantees on the learner-agnostic risk, highlighting the viability of downstream learning in the presence of reasonable learner sets.

## 4.1 LARP for scalar mean estimation

We first analyze LARP for scalar mean estimation under strong contamination (Diakonikolas & Kane, 2023). In this model, an adversary is allowed to inspect $n$ i.i.d. samples from $\mathcal{D}_\theta$ and replace up to an $\epsilon$-fraction of them with arbitrary values. In particular, the initial $\epsilon$-contaminated sample is $S \in \mathbb{R}^n$ with size $n$. The learner set $\mathcal{L}$ consists of Huber estimators (Huber, 1964; Sun et al., 2020; Pensia et al., 2024) parametrized by $\delta > 0$. They are defined as $\hat{\theta}_\delta(S) := \arg\min_{\hat{\theta}} \sum_{x \in S} H_\delta(x - \hat{\theta})$, where the Huber loss $H_\delta$ is defined as

$$H_\delta(x) := \begin{cases} \frac{1}{2}x^2 & \text{for } |x| \leq \delta \\ \delta\left(|x| - \frac{1}{2}\delta\right) & \text{otherwise.} \end{cases}$$

In the cases $\delta \to 0$ and $\delta \to \infty$, $\hat{\theta}_\delta$ converges to the sample median and the sample mean respectively. Intermediate values of $\delta$ represent the trade-off between robustness and sample efficiency. We denote the set of the parameters of the given Huber learners as $\Delta := \{\delta : H_\delta \in \mathcal{L}\}$. Finally, we define our risk function to be the squared distance $R_l(l(S'), \theta) := (l(S') - \theta)^2$.

### 4.1.1 Hardness of learning with high heterogeneity

The heterogeneous behavior of the downstream learners can lead to high losses, no matter the prefiltering. To show this we provide a lower bound on the learner-agnostic risk, by considering a specific example of a target dataset, noise, and learner set.

**Proposition 4.1.** *There is an instance of LARP with specified $\mathcal{D}_\theta$ and fixed $\mathcal{L}$ such that: 1) there exists a prefiltering with $\min_{l \in \mathcal{L}} R_l = \mathcal{O}(\epsilon^2)$, but 2) for all prefiltering procedures, $R_{agn} = \Omega(1)$.*

---

[3]For fixed $R_1 \geq R_2$, the value $\mathcal{U}_{red}(R_1, R_2)$ is the reduction in learner $l$'s utility if their risk is $R_1$ instead of $R_2$.

We refer to Appendix B.1 for proof and more thorough discussion. This lower bound arises due to the presence of learners that are inefficient for the instance. This is a strong indicator that non-trivial bounds must depend on the properties of the learner set $\mathcal{L}$.

### 4.1.2 Quantile-based prefiltering procedure

We now prove feasibility of LARP in the context of scalar mean estimation for a particular family of distributions, which we describe as follows:

**Definition 4.2** (Strict Smoothness). A distribution $\mathcal{D}$ with mean $\mu$ is $(s, \epsilon)$-strictly smooth, where $s = s(\epsilon)$, if

$$\max\left\{\mathbb{P}_{X \sim \mathcal{D}}[X \geq \mu + s], \mathbb{P}_{X \sim \mathcal{D}}[X \leq \mu - s]\right\} < \tfrac{1}{2} - \epsilon$$

This notion allows us to bound the distance between the median of the contaminated distribution and the true mean $\theta$.

*Remark* 4.3. The notion of strict smoothness that we provide in Definition 4.2 is a stronger version of the notion of $(s, \epsilon)$-smoothness provided in Exercise 1.4 of Diakonikolas & Kane (2023). In particular, every distribution that is $(s, \epsilon)$-strictly smooth is also $(s, \epsilon)$-smooth. Furthermore, the two notions are equivalent for all continuous distributions which have positive density over the whole real line.

The prefiltering procedure is inspired by a popular informal rule for marking the tails of a sample as outliers (Maronna et al., 2006). This can be adapted to define the following outlyingness measure:

$$Q\left(x, \{X_1, \ldots, X_n\}\right) = \frac{\left|\min\left\{i : X_{(i)} \geq x\right\} - n/2\right|}{n}.$$

Then, we define a quantile prefiltering procedure as

$$F_p^q(S) := \left\{X \in S : Q(X, S) < p\right\}.$$

The hyperparameter $p$ lies in the range $(0, 1/2 - 1/2n)$. The following result provides guarantees for $R_{agn}(F_p^q)$.

**Theorem 4.4.** *Assume that the target distribution $\mathcal{D}_\theta$ has mean $\theta$ and is $(s(\epsilon), \epsilon)$-strictly smooth for all $\epsilon \in [0, \epsilon_0)$, for some $\epsilon_0 > 0$. Let $F_p^q$ be the quantile prefiltering procedure with any $p \in (0, 1/2 - 1/2n)$. Then, for an $\epsilon$-contaminated sample of sufficiently large size $n$ (with $\epsilon_0 > \epsilon > 0$), with probability $1 - \delta_0$, the prefiltering $F_p^q$ satisfies*

$$R_{agn}(F_p^q) \leq \mathcal{O}\left(s\left(\epsilon + \sqrt{\ln(2/\delta_0)/2n}\right)^2 + \max_{\delta \in \Delta} \delta^2\right). \tag{4}$$

*Proof.* We begin by bounding the distance between the true mean $\theta$ and the median of the contaminated distribution, similarly to Exercise 1.4 of Diakonikolas & Kane (2023). Let $q(r)$ denote the theoretical $r$-quantile of $\mathcal{D}$. More formally, $q(r) := \inf\left\{x \in \mathbb{R} : \mathbb{P}_{X \sim \mathcal{D}}(X \leq x) \geq r\right\}$. Similarly, let $\hat{q}(r)$ be the sample $r$-quantile of the empirical distribution of the clean sample before the contamination. Additionally, let $F$ and $\hat{F}$ be the CDFs of $\mathcal{D}$ and the empirical distribution of the clean sample respectively. The median $m$ of the contaminated sample lies between $\hat{q}(1/2 - \epsilon)$ and $\hat{q}(1/2 + \epsilon)$. By the Dvoretzky-Kiefer-Wolfowitz inequality, with probability $1 - \delta_0$ we have that

$$\sup_{x \in \mathbb{R}}|\hat{F}(x) - F(x)| \leq \epsilon_n := \sqrt{\frac{\ln 2/\delta_0}{2n}}. \tag{5}$$

Now, let $x_0 = q(1/2 + \epsilon + \epsilon_n)$, so $F(x_0) \geq 1/2 + \epsilon + \epsilon_n$. From Eq. (5) we have $|F(x_0) - \hat{F}(x_0)| \leq \epsilon_n$, hence it follows that $\hat{F}(x_0) \geq F(x_0) - \epsilon_n \geq 1/2 + \epsilon$. Going back to the quantiles, we derive $x_0 \geq \hat{q}(1/2 + \epsilon)$. Now, let $x_1 = q(1/2 + \epsilon - \epsilon_n) = \inf\left\{x \in \mathbb{R} : F(x) \geq 1/2 + \epsilon - \epsilon_n\right\}$. Then, for all $x < x_1$ we have $F(x) < 1/2 + \epsilon - \epsilon_n$. But by the DKW inequality we have that for all $x < x_1$ it holds that $\hat{F}(x) \leq F(x) + \epsilon_n < 1/2 + \epsilon$. Hence, we can deduce that $\hat{q}(1/2 + \epsilon) := \inf\{x : \hat{F}(x) \geq 1/2 + \epsilon\} \geq x_1$. Combining the inequalities for $x_0$ and $x_1$ we get $q(1/2 + \epsilon - \epsilon_n) \leq \hat{q}(1/2 + \epsilon) \leq q(1/2 + \epsilon + \epsilon_n)$. In a similar way, we obtain the following inequalities $q(1/2 - \epsilon - \epsilon_n) \leq \hat{q}(1/2 - \epsilon) \leq q(1/2 - \epsilon + \epsilon_n)$. Hence, we have that with probability $1 - \delta_0$ that

$$q\left(1/2 - \epsilon - \epsilon_n\right) \leq m \leq q\left(1/2 + \epsilon + \epsilon_n\right) \tag{6}$$

Now, for sufficiently large $n$, we have that $\epsilon + \epsilon_n < \epsilon_0$ and hence $\mathcal{D}$ is $(s(\epsilon + \epsilon_n), \epsilon + \epsilon_n)$-strictly smooth, so by definition we have

$$\mathbb{P}\left(X \geq \theta + s(\epsilon + \epsilon_n)\right) < \frac{1}{2} - \epsilon - \epsilon_n \quad \text{and} \quad \mathbb{P}\left(X \leq \theta - s(\epsilon + \epsilon_n)\right) < \frac{1}{2} - \epsilon - \epsilon_n.$$

Rewriting this using $q$ we get

$$q\left(1/2 + \epsilon + \epsilon_n\right) \leq \theta + s(\epsilon + \epsilon_n) \quad \text{and} \quad q\left(1/2 - \epsilon - \epsilon_n\right) \geq \theta - s(\epsilon + \epsilon_n).$$

Combining them, we get

$$\theta - s\left(\epsilon + \epsilon_n\right) \leq q\left(1/2 - \epsilon - \epsilon_n\right) \leq m \leq q\left(1/2 + \epsilon + \epsilon_n\right) \leq \theta + s(\epsilon + \epsilon_n).$$

Hence we deduce that

$$|m - \theta| \leq s(\epsilon + \epsilon_n). \tag{7}$$

The quantile mechanism always preserves the median, hence the median $m'$ of the prefiltered sample retains the above property.

We now focus on providing an upper bound on the distance between a Huber estimator $\hat{\theta}_\delta$ and the sample median $m$. First, we note that the Huber estimator minimizes the loss $\sum_{x \in S'} H_\delta(x - \hat{\theta})$. This is equivalent to solving the equation $\sum_{x \in S'} H'_\delta(x - \hat{\theta}) = 0$, where

$$H'_\delta(x) = \begin{cases} x & \text{for } |x| \leq \delta \\ \delta\, sign(x) & \text{otherwise.} \end{cases},$$

where $sign$ denotes the sign function. Note in particular that $H'_\delta(x) \in [-\delta, \delta]$ for all $x \in \mathbb{R}$. Hence, if $\hat{\theta}_\delta - m' > \delta$, we are bound to have a positive value for $\sum_{x \in S'} H'_\delta(x - \hat{\theta}_\delta)$. Similar reasoning goes for the case where $\hat{\theta}_\delta - m' < -\delta$. Hence, for any Huber learner $\hat{\theta}_\delta$ with parameter $\delta$ trained on the prefiltered sample we have the bound $|\hat{\theta}_\delta - m'| \leq \delta$. Combining this bound with Eq. (7) and substituting $\epsilon_n$ we get

$$|\hat{\theta}_\delta - \theta|^2 \leq 2|\hat{\theta}_\delta - m'|^2 + 2|m' - \theta|^2 \leq \mathcal{O}\left(\delta^2 + s\left(\epsilon + \sqrt{\frac{\ln 2/\delta_0}{2n}}\right)^2\right).$$

This gives us the final upper bound on the worst-case estimator

$$R_{agn}(F_p^q) = \max_{\delta \in \Delta}|\hat{\theta}_\delta - \theta|^2 \leq \mathcal{O}\left(\max_{\delta \in \Delta} \delta^2 + s\left(\epsilon + \sqrt{\frac{\ln 2/\delta_0}{2n}}\right)^2\right).$$

$\square$

The first term in Eq. (4) is reminiscent of upper bounds in robust mean estimation (Diakonikolas & Kane, 2023) and increases with the corruption rate $\epsilon$ and decreases with the sample size. The second term depends solely on the learner set and is always nonnegative. More concretely, small values of $\delta \approx 0$ yield estimators close to the sample median, which is known to enjoy guarantees similar to the first term in Eq. (4) (Diakonikolas & Kane, 2023). Therefore, if all estimators in $\mathcal{L}$ use a small value of $\delta$ (i.e., all estimators are "good" for the considered learning problem), the bound is comparable to those in standard robust mean estimation. However, if some estimators use a large $\delta$, they are suboptimal for the problem and the bound is also larger as $\max_{\delta \in \Delta} \delta^2$ is large. This aligns with the arguments presented in Section 4.1.1. In Appendix D.1, we empirically evaluate the effectiveness of the quantile-based prefiltering mechanism, alongside several alternative approaches.

If the target distribution $\mathcal{D}$ is Gaussian, our upper bound takes the following form:

**Corollary 4.5.** *In the setup of Theorem 4.4, if $\mathcal{D}_\theta = \mathcal{N}(\theta, \sigma^2)$ and $\epsilon_0 < 2/7$, we have*

$$R_{agn}(F_p^q) \leq \mathcal{O}\left(\left(\epsilon^2 + \ln(1/\delta_0)/n\right)\sigma^2 + \max_{\delta \in \Delta} \delta^2\right).$$

This assumption on the target distribution allows us to further quantify the behavior of the upper bound on the learner-agnostic risk. We present the proof of this result in Appendix B.2.

## 4.2 LARP with oracle prefiltering procedures

In order to strengthen the evidence for the viability of LARP beyond scalar mean estimation, we introduce another theoretical setup which is concerned with idealized representations of learners and prefiltering procedures. We show how robustness guarantees on the individual learner performance can be used to derive guarantees on the LARP objective, which also depend on the effectiveness of the prefiltering procedure.

**Setup** We introduce the concept of an oracle prefiltering procedure $F_p^{o,t}$, parametrized by $p \in [0,1)$ denoting what fraction of data is removed, and a new parameter $t \in [0,1]$ describing its effectiveness. We model $F_p^{o,t}$ as a function which independently selects whether to remove or keep each point, using the ground truth knowledge of whether it is noisy or not. The new parameter $t$ measures what fraction of the removed points are noisy, up to the point when all noisy data is removed. Hence, after prefiltering with $F_p^{o,t}$ we are left with a sample of size $(1-p)n$ in which $\max\{(\epsilon - tp)n, 0\}$ points are contaminated. In practice $t$ is a property of the prefiltering algorithm while $p$ is a hyperparameter which may be tuned according to the setup. We analyze $F_p^{o,t}$ theoretically, in the context of PAC learning. Taking inspiration from classic upper bounds (Kearns & Li, 1988), we model the learners $l \in \mathcal{L}$ as algorithms that take a sample of size $n_0$ and contamination rate $\epsilon_0$ and return hypotheses $h_{\epsilon_0, n_0}^l$ such that

$$R_l(h_{\epsilon_0, n_0}^l, \theta) \le A_l \epsilon_0 + B_l / \sqrt{n_0},$$

uniformly over all $l \in \mathcal{L}$, with probability $1 - \delta(n)$, where $\delta(n) \to 0$ as $n \to \infty$. The different values of $A_l$ and $B_l$ represent the different amounts of robustness and statistical efficiency present in each learner. In this setup we show bounds on the learner-agnostic risk.

**Theorem 4.6.** *Given an oracle prefiltering procedure with $t > \epsilon$, there is a value $p$ such that, with probability $1 - \delta(n)$, $F_p^{o,t}$ satisfies*

$$R_{agn}(F_p^{o,t}) \le \min\left\{ \max_{l \in \mathcal{L}} \left[ A_l t + \frac{1}{4n(t - \epsilon)} \frac{B_l^2}{A_l} \right], \frac{\max_{l \in \mathcal{L}} B_l}{\sqrt{n(t - \epsilon)}}, \max_{l \in \mathcal{L}} \left[ A_l \epsilon + \frac{B_l}{\sqrt{n}} \right] \right\}. \tag{8}$$

*Proof.* By the definition of the learner-agnostic risk we have

$$R_{agn}(F_p^{o,t}) = \max_{l \in \mathcal{L}} R_l\left( F_p^{o,t} \right)$$

But after prefiltering with $F_p^{o,t}$, each learner gets a sample with size $n(1-p)$ and contamination rate $\max\{(\epsilon - tp), 0\}/(1-p)$, hence the risk of each learner $l \in \mathcal{L}$ enjoys the bound

$$R_l(F_p^{o,t}) \le \frac{A_l \max\{(\epsilon - tp), 0\}}{1-p} + \frac{B_l}{\sqrt{n(1-p)}}$$

uniformly over $\mathcal{L}$, with probability $1 - \delta(n)$. Taking the maximum over $\mathcal{L}$, we get

$$R_{agn}(F_p^{o,t}) \le R_{agn}'\left( F_p^{o,t} \right) := \max_{l \in \mathcal{L}} \left[ \frac{A_l \max\{(\epsilon - tp), 0\}}{1-p} + \frac{B_l}{\sqrt{n(1-p)}} \right]$$

with probability $1 - \delta(n)$. From this point on, we aim to provide an upper bound on $R_{agn}'$. First, it is easy to observe that $R_{agn}'$ is increasing for $p \in [\epsilon/t, 1)$, hence we can restrict $p$ to the interval $[0, \epsilon/t]$. In this interval, $R_{agn}(F_p^{o,t})$ achieves its minimum over $p$, so we are interested in bounding from above

$$R'' := \min_{p \in [0, \epsilon/t]} \max_{l \in \mathcal{L}} \left[ \frac{A_l(\epsilon - tp)}{1-p} + \frac{B_l}{\sqrt{n(1-p)}} \right]$$

Let us write $u := 1 - p$, so we can write the learner-agnostic risk as

$$R'' = \min_{u \in [1 - \epsilon/t, 1]} \max_{l \in \mathcal{L}} \left[ \frac{A_l(\epsilon - t(1-u))}{u} + \frac{B_l}{\sqrt{nu}} \right] = \min_{u \in [1 - \epsilon/t, 1]} \max_{l \in \mathcal{L}} \left[ A_l t - \frac{A_l(t - \epsilon)}{u} + \frac{B_l}{\sqrt{nu}} \right].$$

Let $f_l(u) = A_l t - A_l(t - \epsilon)/u + B_l/\sqrt{nu}$. For each $l \in \mathcal{L}$, we first calculate the derivative of $f_l$ as $f_l'(u) = A_l(t - \epsilon)u^{-2} - B_l n^{-1/2} u^{-3/2}$. Hence $f_l$ has a unique critical point at $u_l^* := 4nA_l^2(t - \epsilon)^2/B_l^2$. Moreover, the derivative is strictly positive for $u < u_l^*$ and strictly negative for $u > u_l^*$, hence $u_l^*$ is the unique maximum of $f_l$ for $u > 0$. Hence, for all $u \in [1 - \epsilon/t, 1]$ we have

$$f_l(u) \le f_l(u_l^*) = A_l t + \frac{B_l^2}{4nA_l(t - \epsilon)}.$$

Hence we can give the following bound on the minimax:

$$R'' = \min_{u \in [1 - \epsilon/t, 1]} \max_{l \in \mathcal{L}} \left[ A_l t - \frac{A_l(t - \epsilon)}{u} + \frac{B_l}{\sqrt{nu}} \right] \le \max_{l \in \mathcal{L}} \left[ A_l t + \frac{1}{4n(t - \epsilon)} \frac{B_l^2}{A_l} \right].$$

Furthermore, we can provide an additional upper bound on the minimax value by fixing the values $u_1 = 1$ and $u_2 = 1 - \epsilon/t$.

$$R'' \le \max_{l \in \mathcal{L}} f_l(u_1) = \max_{l \in \mathcal{L}} \left[ A_l \epsilon + \frac{B_l}{\sqrt{n}} \right] \quad \text{and} \quad R'' \le \max_{l \in \mathcal{L}} f_l(u_2) = \frac{\max_{l \in \mathcal{L}} B_l}{\sqrt{n(t - \epsilon)}}.$$

Combining the three upper bounds on $R''$, we get the desired inequality, since for $p$ which achieves the minimum we have $R_{agn}\left(F_p^{o,t}\right) \le R_{agn}'\left(F_p^{o,t}\right) = R''$ with probability $1 - \delta(n)$. □

The assumption $t > \epsilon$ is reasonable as it signifies that prefiltering is useful in the sense that it reduces contamination as it prefilters data. Writing the right-hand side in Eq. (8) in such a way allows us to choose whichever of the upper bounds is better for particular learner parameters. In particular, we note that for $\epsilon = 0$ we recover the standard $1/\sqrt{n}$ rate.

## 5 Price of learner-agnostic prefiltering

Previous results highlight an inherent trade-off when performing LARP: protecting a broader set of learners via the single prefiltering leads to more pessimistic accuracy guarantees compared to learner-specific instances. This reduction can be measured through the notion of price of learner-agnostic prefiltering as defined in Section 3. In this section we measure the price of learner-agnostic prefiltering in different realistic empirical setups. We conduct parametric analysis to investigate the dependence on contamination rate, learner heterogeneity and dataset size. We also measure the price of LARP in a modification of our framework which considers learners with heterogeneous downstream risk functions. Finally, we study a game-theoretic model in which downstream learners can split the cost of prefiltering, providing an additional theoretical argument for the benefits of LARP on large datasets from the perspective of saving on total data curation costs.

### 5.1 Price of LARP on real-world data

In this section, we quantify the price of learner-agnostic prefiltering on datasets with inherent noise. Specifically, we conduct experiments in both the tabular and image classification settings in the presence of uniform label noise. In Appendix D we provide modifications of our setup which use different prefiltering procedures, data contamination or learner sets. In all cases, the price $P$ is computed using $\mathcal{U}_{red}(R_1, R_2) := (R_1 - R_2)/R_2$, which measures the relative increase in classification risk when learner-agnostic prefiltering is used.

**Datasets** We consider several standard ML benchmarks. For each of them, we reserve a part of the dataset as risk evaluation data. This is uncorrupted data that we use to evaluate the quality of the resulting learners, which is never seen by the data provider and/or downstream learners. The remaining dataset corresponds to the dataset $S$ in our model, which gets corrupted and then prefiltered and fed as input to the downstream learners. Whenever the learners use validation data to set hyperparameters, this validation data is a subset of the prefiltered dataset $F(S)$.

For image experiments we use CIFAR-10 (Krizhevsky, 2009), using the train set as the dataset $S$ and the test set as risk evaluation data. Tabular experiments utilize Adult (Becker & Kohavi, 1996), randomly split

into the data $S$ (80%) and risk evaluation data (20%). We apply $\epsilon = 30\%$ uniform label noise to $S$ and use the clean risk evaluation data to measure downstream learner generalization. For CIFAR-10, the risk metric $R$ is classification error rate, whereas for Adult we use macro-F1 score (Manning et al., 2008) to account for class imbalance. In Appendix D we provide additional experiments with **i)** CIFAR-10 with shortcuts, **ii)** CIFAR-10N dataset (Wei et al., 2022), which contains human annotation label noise, and **iii)** Tiny ImageNet dataset (Wu et al., 2017) with label noise.

**Prefiltering procedures** In Algorithm 1 we present a general description of the prefiltering procedures we use throughout our experiments. Given the dataset $S$, the algorithm first trains a scoring model $\mathcal{M}_\theta$ (e.g., ResNet-9 (He et al., 2016) on CIFAR-10, and two-layer fully-connected NN on Adult, for our label noise experiments). Then, Algorithm 1 executes a scoring function $g$, assigning a score to each sample in $S$. For example, in the following experiments on label noise, we set $g(z, \mathcal{M}_\theta)$ as the loss of $\mathcal{M}_\theta$ on the particular data point $z$. This is motivated by the fact that neural networks tend to learn useful features before overfitting to label noise (Zhang et al., 2017; Arpit et al., 2017), hence noisy points tend to have higher losses during early training iterations. Then, the algorithm gathers all the scores, computes a threshold $\tau$ as the $(1-p)$-th quantile of all scores, and filters out all points above that quantile threshold. Finally, the algorithm splits the prefiltered dataset $S'$ into train and validation sets, the latter being utilized by the learners for early-stopping. In Appendix E we discuss how Algorithm 1 captures our empirical setups, as well as real-world data curation strategies. We found this procedure effective in practice. In our setting with $\epsilon = 30\%$, a removal fraction of $p = 25\%$ successfully filtered $> 75\%$ of the corrupted samples. We also study a prefiltering procedure based on the Confident Learning (CL) approach (Northcutt et al., 2021a) described in Appendix D, as well as oracle prefiltering as described in Section 4.2. All prefiltering procedures are parametrized by the fraction $p$ of the dataset $S$ that is being prefiltered.

---

**Algorithm 1** General Prefiltering Procedure

---

**Require:** Provider Dataset $S$, fraction $p \in [0, 1)$, scoring function $g(\cdot, \cdot)$, and scoring model/estimator $\mathcal{M}$
1: $\mathcal{M}_\theta \leftarrow \text{Fit}(S, \mathcal{M})$       // Fit the scoring model
2: $Scores \leftarrow \emptyset$
3: **for** $z_i \in S$ **do**
4:     $s_i \leftarrow g(z_i, \mathcal{M}_\theta)$       // Compute outlyingness (e.g., loss or z-score)
5:     $Scores.\text{append}(s_i)$
6: **end for**
7: $\tau \leftarrow \text{Quantile}(Scores, 1-p)$       // Find the cutoff threshold
8: $S' \leftarrow \{z_i \in S \mid g(z_i, \mathcal{M}_\theta) < \tau\}$       // Keep only points below threshold
9: $S'_{train}, S'_{val} \leftarrow \text{TrainValSplit}(S')$       // Split into train and validation sets
10: **return** $S'_{train}, S'_{val}$

---

**Learner sets** We evaluate LARP on sets of models with varying amounts of robustness and statistical efficiency. As we train a large number of models, for CIFAR-10 we opt to use convolutional networks (CNNs) with an architecture (see Appendix C.1) optimized for fast training that still maintains good accuracy. As we observe that L2 regularization increases the robustness to noisy labels in practice, but is also susceptible to underfitting, we generate our set of 8 models $\mathcal{L}$ by varying the CNNs' L2 regularization parameter in the range $[3e-3, 2e-2]$. For Adult experiments, we opt to use XGBoost (Chen & Guestrin, 2016), AdaBoost (Freund & Schapire, 1997), LogitBoost (Friedman et al., 2000), Bagging (Breiman, 1996), RandomForest (Breiman, 2001), SVM (Cortes & Vapnik, 1995), as well as neural networks with two hidden layers and ReLU activations; in order to show how the price of LARP behaves under a diverse set of machine learning model types.

We train all neural networks using PyTorch (Paszke et al., 2019) with the Adam optimizer (Kingma & Ba, 2015) and the remaining models are trained using Scikit-learn (Pedregosa et al., 2011). Full experimental details can be found in Appendix C.1.

**Results** We begin by confirming that the prefiltering procedure we consider can improve learning performance despite label noise, by studying its effectiveness in the learner-specific setting. In Fig. 1 we see that

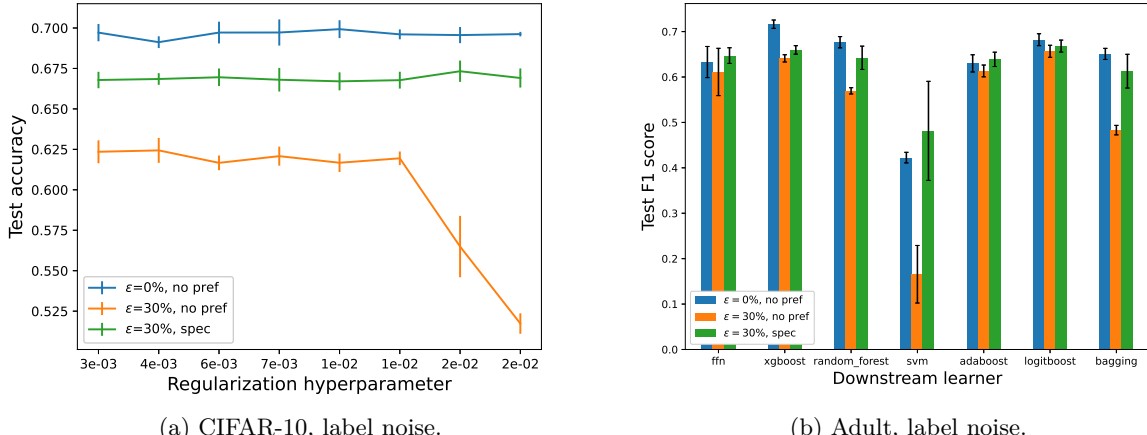

(a) CIFAR-10, label noise.  (b) Adult, label noise.

Figure 1: Performance of learners in the absence of label noise (blue), in the presence of noise but no prefiltering (orange), and in the presence of prefiltering in the learner-specific regime (green).

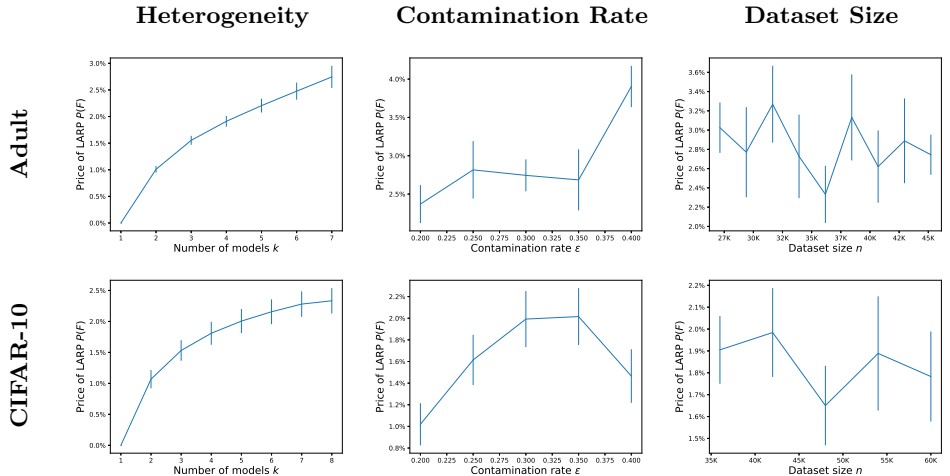

Figure 2: Price of learner-agnostic prefiltering for Adult (upper row) and CIFAR-10 (lower row). Each column corresponds to a different hyperparameter of the setup we measure the price against.

the prefiltering procedures bring positive impact to downstream models. This shows that the prefiltering procedures we use are effective for most downstream learners. Most notably, downstream CNNs yield between 52% and 62.5% accuracy in the presence of noise and no prefiltering, whereas generalization accuracy increases to 67% in the presence of prefiltering that is individually tailored to them. On the other hand, the learners in our experiment with tabular data show different improvements in F1 score in the presence of prefiltering, from little improvement (e.g., FFN shows no statistically significant improvement, remaining at 0.63) to significant increases (e.g., SVM improves its average F1 score from 0.17 to 0.48).

The above results serve as an indication that the prefiltering procedures can improve the downstream performance of (most) learners. However, providing simultaneous guarantees on the entire learner set is more difficult, as captured by the price of LARP $P(F)$. In Fig. 2 we show how $P(F)$ is affected by the different factors in the equation of the learner-agnostic risk $R_{agn}$ in Theorem 4.4. In particular, we show the effects of the contamination rate $\epsilon$, the dataset size $n$, and the learner heterogeneity, expressed in terms of the average price $P$ on all subsets of size $k$ of the learner set $\mathcal{L}$.

Table 1: Price of LARP $P(F)$ (mean and s.e., in %) for default, oracle (by effectiveness $t$), and CL-based prefiltering procedures. The rows describe the dataset and whether the learner sets have low or high heterogeneity.

| | Default | Oracle | | | | | | CL-based |
|---|---|---|---|---|---|---|---|---|
| | | t=65% | t=70% | t=75% | t=80% | t=85% | t=90% | |
| Adult(low) | 1.2(0.4) | 1.7(0.2) | 1.4(0.2) | 1.3(0.2) | 1.1(0.2) | 1.2(0.1) | 1.1(0.2) | 2.0(0.1) |
| Adult(high) | 1.4(0.4) | 2.2(0.3) | 1.7(0.3) | 1.8(0.3) | 1.4(0.3) | 1.6(0.2) | 1.4(0.3) | 2.7(0.2) |
| CIFAR-10(low) | 1.5(0.1) | 1.0(0.1) | 1.7(0.2) | 1.5(0.2) | 1.1(0.2) | 1.4(0.1) | 1.4(0.2) | 1.3(0.1) |
| CIFAR-10(high) | 2.2(0.2) | 1.4(0.2) | 2.4(0.2) | 2.3(0.2) | 1.7(0.3) | 2.1(0.2) | 2.0(0.4) | 2.0(0.2) |

Crucially, we see a statistically significant price of learner-agnostic prefiltering $P$. This is in line with our results in Theorem 4.4. Our results are consistent across setups, suggesting that the price of LARP can be incurred for different data modalities and learner sets.

We observe that heterogeneity has the most pronounced effect on the price of LARP — as $k$ and hence heterogeneity increases, the price increases, starting from $k = 1$ where $P(F) = 0$ by definition. In Appendix D, we study a different notion of learner heterogeneity — learner diameter, which measures the range of regularization parameters used by the set of learners $\mathcal{L}$, again confirming the heterogeneity impact on $P$.

Another important factor for the price $P$ is the contamination rate $\epsilon$. In our experiments, we observe that in general $P$ increases together with $\epsilon$. However, there are settings where too large $\epsilon$ leads to a decrease in $P$. In general, we observe that $P(F)$ attains its lowest values when $\epsilon$ is very small or very large. We believe the reason for this is that in the former case all learners prefer the values $p$ which result in the contaminated data being almost perfectly prefiltered, whereas in the latter case all learners prefer little to no prefiltering.

Finally, the effect of the dataset size $n$ on $P(F)$ appears to be weak. This hints at increasing benefits of LARP as we scale $n$ since the accuracy drop can be offset by the growing cost of conducting a prefiltering procedure. We model and study this offset in detail in Section 5.3.

We also show a consistently significant signal of $P(F)$ across different prefiltering procedures. In Table 1 we present $P(F)$ for our default, oracle and CL-based prefiltering procedures. We show results in two settings, one with $k = 2$ learners (low heterogeneity), and one with all learners (high heterogeneity). In Appendix D we expand upon the results of Table 1, showing the dependence of $P(F)$ on learner heterogeneity, dataset size, contamination rate, and effectiveness for the oracle and CL-based (Northcutt et al., 2021a) methods.

First, we observe that the price of LARP for both the default and the CL-based prefiltering procedures behaves similarly to the oracle instances. This confirms the validity of the former two prefiltering procedures as good proxies for the idealized, yet impractical, oracles. Furthermore, oracle prefiltering procedures allow us to study $P(F)$ as a function of the effectiveness of the prefiltering. In particular, we see that $P$ attains its lowest values when $t$ is either small or close to 1. This is in line with previous analysis of $P(F)$ as a function of $\epsilon$, as we generally observe an increase in $P(F)$ but we also observe that for large values of $t$ the price of LARP decreases.

## 5.2 Price of LARP for heterogeneous risks

In the previous subsection we explored an instance of LARP in which the heterogeneity in the learner set stems from the difference in the learning algorithms. However, it is often the case that the different downstream learners use the same learning algorithm, yet their risk functions differ. In this subsection we provide a modification of LARP in which the learners differ only in their risks. We consider a binary classification task in which learners balance accuracy and fairness.

**Setup** The unfiltered dataset is the base variant of the Bank Account Fraud (BAF) dataset suite (Jesus et al., 2022), which we split into train (80%) and risk evaluation (20%) sets. This dataset is doubly imbalanced with respect to both **i)** label distribution, and **ii)** the sensitive attribute. Our prefiltering procedure is the

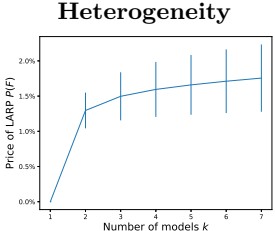 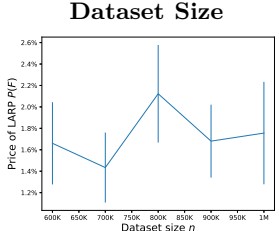

Figure 3: Price of learner-agnostic prefiltering for fair binary classification on the BAF dataset.

algorithm from Yalcin et al. (2025), which filters out data with specific proportions to alleviate such imbalance. We track the Matthews Correlation Coefficient (MCC) (Chicco & Jurman, 2020), measuring model accuracy, and Disparate Impact rate (DI) (Feldman et al., 2015), measuring a model's fairness, and we define the risk as

$$R_l := a_l|1 - MCC| + b_l|1 - DI|.$$

All 7 learners use the same model (Random Forest with n_estimators=200), but their risk metrics have $a_l = 1$ and differ in $b_l \in [0, 2]$. This reflects the difficulty of a single prefiltering procedure to cater to conflicting objectives, even when the model is the same. In Fig. 3 we show the effect of dataset size $n$ and heterogeneity (measured using $k$) on the price of LARP $P(F)$ ($\epsilon$ is not applicable to this setting), observing similar trends to the setup in Section 5.1. In particular, the prevalent weak dependence between $P(F)$ and $n$ motivates us to explore the practical benefits of LARP on large datasets in Section 5.3.

### 5.3 Benefits of learner-agnostic prefiltering

In all prior results, we observe that while LARP can protect multiple downstream learners, model performance is usually more pessimistic than when learner-specific prefiltering is applied. At the same time, prefiltering for an individual use-case can be costly for a dataset user, due to, e.g., computational costs and expert hours.

In this subsection we study a game-theoretic model where downstream learners can split the cost of the single prefiltering under LARP, enabling a comparison between the price of LARP and savings in repeated data curation efforts. In our model, each learner can either choose learner-specific prefiltering and pay the full cost of preprocessing the dataset, or opt for learner-agnostic prefiltering and split said cost with other learners. We provide sufficient conditions under which all learners are provably incentivized to participate in LARP.

**Definition** Within the framework of LARP, we consider a game in which each learner $l \in \mathcal{L}$ is a player maximizing their utility. Each learner $l$ has utility $U^l$ that is a function of their risk $R_l(F)$, as well as the cost they pay for the prefiltering procedure.

We model the total cost of prefiltering a dataset of size $n$ as $Cn^\alpha$ with $C > 0, \alpha \geq 1$. The constant $\alpha$ describes the complexity of conducting the prefiltering procedure, and $\alpha \geq 1$ is reasonable since each data point needs to be processed, which already induces linear complexity.

Each learner selects an action $a_l \in \{0, 1\}$ indicating if they prefer a learner-specific or learner-agnostic procedure. If learner $l$ picks $a_l = 0$, they receive individual prefiltering, which costs $Cn^\alpha$, and receive risk $R_l(F_l^*)$, yielding final utility $U_{spec}^l$. If learner $l$ plays $a_l = 1$, then they participate in learner-agnostic prefiltering with other players. Then, all learners split the cost $Cn^\alpha$ of prefiltering according to a vector $(p_l)_{l \in \mathcal{L}}$, i.e., $\sum_{l \in \mathcal{L}} p_l = Cn^\alpha$. Then learner $l$ loses $\mathcal{U}_{red}(R_l(F), R_l(F_l^*))$ utility from reduced performance, but they lower their cost of prefiltering from $Cn^\alpha$ to $p_l$. The utility for player $l$ becomes $U_{agn}^l = U_{spec}^l - \mathcal{U}_{red}(R_l(F), R_l(F_l^*)) + Cn^\alpha - p_l$. These utilities connect directly to the price $P(F)$ defined in Eq. (3).

We provide a sufficient condition for the utility benefit of LARP over learner-specific prefiltering.

**Theorem 5.1.** *Assume that in the aforementioned setup, the dataset size $n$ satisfies*

$$n > \left[\frac{|\mathcal{L}|}{C(|\mathcal{L}| - 1)} P(F)\right]^{1/\alpha}. \tag{9}$$

*Then, there is a payment scheme* $(p_l)_{l \in \mathcal{L}}$ *such that* $U_{agn}^l \geq U_{spec}^l$ *for all* $l \in \mathcal{L}$. *In other words, no learner is incentivized to opt out of the learner-agnostic prefiltering scheme.*

*Proof.* Suppose that the condition is satisfied. Let us pick a cost distribution $(p_l)_{l \in \mathcal{L}}$ such that

$$p_l \propto Cn^\alpha - \mathcal{U}_{red}(R_l(F), R_l(F_l^*))$$

and $\sum_{l \in \mathcal{L}} p_l = Cn^\alpha$. As noted in Section 5.3, we have the difference in utilities between learner-agnostic and learner-specific prefiltering for learner $l$ is given by

$$U_{agn}^l - U_{spec}^l = Cn^\alpha - p_l - \mathcal{U}_{red}(R_l(F), R_l(F_l^*)).$$

On the other hand, we have for all $l \in \mathcal{L}$ the bound

$$\sum_{l \in \mathcal{L}} [Cn^\alpha - \mathcal{U}_{red}(R_l(F), R_l(F_l^*))] = |\mathcal{L}|Cn^\alpha - |\mathcal{L}|P(F) \geq |\mathcal{L}|Cn^\alpha - C(|\mathcal{L}| - 1)n^\alpha = Cn^\alpha.$$

Hence, we have for all $l \in \mathcal{L}$ the inequality

$$p_l = \frac{Cn^\alpha - \mathcal{U}_{red}(R_l(F), R_l(F_l^*))}{\sum_{l' \in \mathcal{L}} [Cn^\alpha - \mathcal{U}_{red}(R_{l'}(F), R_{l'}(F_{l'}^*))]} Cn^\alpha \leq Cn^\alpha - \mathcal{U}_{red}(R_l(F), R_l(F_l^*)).$$

This implies that $U_{agn}^l \geq U_{spec}^l$ for all $l \in \mathcal{L}$, as desired. $\qquad\square$

Note that since $P(F)$ may depend on $n$, Eq. (9) need not trivially hold for sufficiently large $n$. However, the next result gives sufficient conditions on $\mathcal{U}_{red}$ and $R_{agn}$.

**Corollary 5.2.** *In the context of Theorem 5.1, assume that either of the following two conditions holds:*

    *i)* $\mathcal{U}_{red}$ *is bounded,*

    *ii) There exists* $L > 0$ *such that* $\mathcal{U}_{red}(x, y) \leq L|x - y|$ *for all* $x \geq y \geq 0$ *and* $R_{agn}/n^\alpha \xrightarrow{\mathbb{P}} 0$ *as* $n \to \infty$.

*Then, for sufficiently large $n$, Eq. (9) holds with high probability over all randomness.*

We refer to Appendix F for discussion on why these assumptions are reasonable, and we present the proof of Corollary 5.2 in Appendix B.3. Furthermore, we observe indications of sublinear growth across all experiments, and Eq. (9) holds for sufficiently large $n$ if $P(F)$ grows as $o(n^\alpha)$. This gives us empirical and theoretical evidence for the overall benefit of LARP from the perspective of saving on data prefiltering costs for downstream dataset users, within the game-theoretic model of this section.

## 6 Conclusion

In this work, we studied the problem of prefiltering public data with learner-agnostic guarantees. We presented a formal framework for the problem and contrasted it with classic robust learning. We used a mean estimation instance of the framework to argue that LARP guarantees should depend on the learner set. We proved upper bounds on the maximum risk in two theoretical setups. We also conducted an empirical analysis of the "price of LARP" on real-world tabular and image data. Finally, we argued for the benefits of LARP for large datasets, from the perspective of saving on costs from repeated data prefiltering by individual dataset users.

We see our work as an initial step towards understanding how prefiltering datasets can provide reliability guarantees for specific downstream inference and learning procedures. In particular, LARP is one way of formalizing the problem of principled data prefiltering by the data provider, with the goal of making the dataset more suitable for downstream ML training. Our theoretical and empirical results signal that it is possible to provide meaningful LARP guarantees and improvements in empirical performance across several natural families of learners. Additionally, in Section 5.3 we argued that LARP can also be beneficial from the perspective of saving on data prefiltering costs of individual dataset users.

Several exciting directions remain for future work. First is the generalization of our theoretical results to regression and classification tasks, as well as the design of provably optimal prefiltering procedures. Second, extending LARP to settings where data preprocessing goes beyond sample removal (e.g., by allowing sample modification) is a natural next step. This is particularly relevant in language modeling, where individual documents may contain both useful and noisy tokens, and a practitioner may prefer to transform a sample rather than discard it entirely. Accommodating such modifications within the LARP framework would require modeling the space of possible edits and the rules for selecting the optimal one, which we leave as an exciting direction for future work.

### Acknowledgments

This research was partially funded by the Ministry of Education and Science of Bulgaria (support for INSAIT, part of the Bulgarian National Roadmap for Research Infrastructure). This project was supported with computational resources provided by Google Cloud Platform (GCP).

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

## A    Outline of supplementary material

The supplementary material is structured as follows.

- Appendix B contains proofs of theoretical results.

- Appendix C contains further experimental details.

- Appendix D contains additional experiments on real-world data.

- Appendix E contains further discussion on Algorithm 1.

- Appendix F contains further discussion of results presented in Corollary 5.2.

- Appendix G contains miscellaneous information.

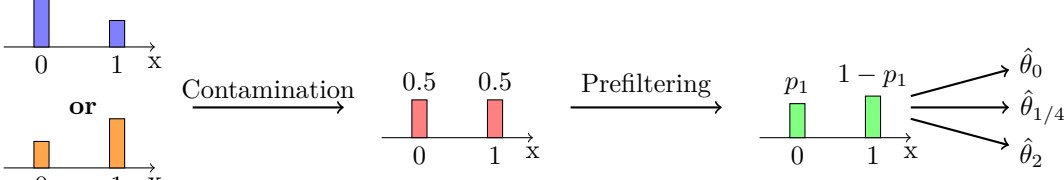

Figure 4: Visual representation of the instance.

# B  Proofs of theoretical results

This section contains the proofs of all theoretical results, and is structured as follows:

- Appendix B.1 contains the proof of Proposition 4.1, as well as further discussion.

- Appendix B.2 contains the proof of Corollary 4.5.

- Appendix B.3 contains the proof of Corollary 5.2.

## B.1  Proof and discussion of Proposition 4.1

We first restate and prove Proposition 4.1.

**Proposition 4.1.** *There is an instance of LARP with specified $\mathcal{D}_\theta$ and fixed $\mathcal{L}$ such that: 1) there exists a prefiltering with $\min_{l \in \mathcal{L}} R_l = \mathcal{O}(\epsilon^2)$, but 2) for all prefiltering procedures, $R_{agn} = \Omega(1)$.*

*Proof.* Recall that the prefiltering procedure aims to minimize the loss presented in Eq. (1). Note that for a finite sample size the prefiltering procedure induces a distribution on the filtered dataset. By letting the sample size go to infinity, this will lead to a limit distribution, which we will denote as $F(P'_\theta)$. We will refer to this limit regime as "infinite sample size". We show an example that has high risk even in the infinite sample size regime.

Assume that the target distribution is $Ber(\theta)$, where either $\theta = (1 - \epsilon)/2$ or $\theta = (1 + \epsilon)/2$, where $\epsilon$ is the contamination rate. Furthermore, assume that the contamination is such that the contaminated distribution is $P'_\theta = Ber(1/2)$. Finally, assume that the downstream learners are Huber estimators with parameters $\delta = 0, \delta = 1/4, \delta = 2$. The first one corresponds to the mean absolute loss minimizer, the last one corresponds to the mean[4], and the middle one corresponds to some intermediate estimator. Let us denote the corresponding estimates by $\hat{\theta}_0, \hat{\theta}_{1/4}$ and $\hat{\theta}_2$ respectively. This instance of the framework is presented in Fig. 4.

Then, any prefiltering procedure $F$ receives the contaminated distribution $P'_\theta$ and maps it to a distribution $F(P'_\theta)$. Moreover, since the procedure is only allowed to filter points, the probability mass function $p'$ of $P'_\theta$ must be absolutely continuous w.r.t. the probability mass function $p$ of $P_\theta$. In particular, this means that $P'_\theta$ is also Bernoulli distributed with some parameter $p_1$. We can describe each possible $F$ with the parameter $p_1 \in [0, 1]$ in the prefiltered distribution. Then, we can explicitly calculate the values of the estimators as a function of $p_1$. In particular, if $p_1 > 1/2$, then

$$\hat{\theta}_0 = 1, \hat{\theta}_{1/4} = 1 - \frac{1 - p_1}{4p_1}, \hat{\theta}_2 = p_1.$$

In the other case, $p_1 \leq 1/2$, then

$$\hat{\theta}_0 = 0, \hat{\theta}_{1/4} = \frac{p_1}{4(1 - p_1)}, \hat{\theta}_2 = p_1.$$

---
[4]Due to the support of the particular distributions, each data point will be considered with squared loss when $\delta = 2$.

We begin by noting that $\min_l R_l(F)$ can be shown to be $\mathcal{O}(\epsilon^2)$, by selecting a prefiltering mechanism that balances out the mass at 0 and 1, i.e., returning the distribution $Ber(1/2)$, and then considering the sample mean estimator $\hat{\theta}_2$.

Nevertheless, the learner-agnostic risk $R_{agn}$ that we aim to minimize satisfies

$$R_{agn}(p_1) = \max_{\delta \in \{0,1/4,2\}} |\hat{\theta}_\delta - \theta|^2 \geq \Omega(1)$$

for all $p_1 \in [0,1]$. $\qquad \square$

*Remark* B.1. The reason for this difference is the existence of estimators which can be considered to be suboptimal for the given task and distribution, and also the difference between the estimates that are given here. Returning to the decomposition presented in Eq. (2), we can see that each of the terms can be made small at the expense of the other. For example, if we pick $p_1 = 1/2$, then we have

$$\min_{\delta \in \{0,1/4,2\}} |\hat{\theta}_\delta - \theta|^2 \leq \mathcal{O}(\epsilon^2).$$

Nevertheless, the second term, corresponding to learner heterogeneity, satisfies

$$\max_{\delta \in \{0,1/4,2\}} |\hat{\theta}_\delta - \theta|^2 - \min_{\delta \in \{0,1/4,2\}} |\hat{\theta}_\delta - \theta|^2 \geq \Omega(1).$$

On the other hand, the last term in Eq. (2) is not bounded from below by any positive constant. Indeed, it can always be reduced to zero by using a prefiltering procedure that collapses the distribution to some delta function. In this case all reasonable estimates will return that constant data point as an estimate that passes through $F$. Of course, in the general case this is not useful since a constant prefiltered distribution leads to huge loss of information about the initial $\theta$. Harking back to our example, if we select $p_1 = 0$ or $p_1 = 1$, then we will achieve zero learner heterogeneity, i.e.,

$$\max_{\delta \in \{0,1/4,2\}} |\hat{\theta}_\delta - \theta|^2 - \min_{\delta \in \{0,1/4,2\}} |\hat{\theta}_\delta - \theta|^2 = 0.$$

Nevertheless, this results in the fact that all estimates are bad in the sense that

$$\min_{\delta \in \{0,1/4,2\}} |\hat{\theta}_\delta - \theta|^2 = \Omega(1).$$

The conclusion that we take from this example is that all meaningful bounds of the learner-agnostic framework must take into account not only the robustness properties of the statistical task, but also the downstream learners, since if they are sufficiently inefficient or non-robust, learning is simply not possible.

## B.2 Proof of Corollary 4.5

**Corollary 4.5.** *In the setup of Theorem 4.4, if $\mathcal{D}_\theta = \mathcal{N}(\theta, \sigma^2)$ and $\epsilon_0 < 2/7$, we have*

$$R_{agn}(F_p^q) \leq \mathcal{O}\left(\left(\epsilon^2 + \ln(1/\delta_0)/n\right)\sigma^2 + \max_{\delta \in \Delta} \delta^2\right).$$

*Proof.* Assume that $\epsilon > 0$. We will show that $\mathcal{D}$ is $(\mathcal{O}(\sigma\epsilon), \epsilon)$-strictly smooth for $\epsilon < 2/7$. Indeed, let $Z \sim \mathcal{N}(0,1)$ and let $C = 4$. Then

$$\mathbb{P}_{X \sim \mathcal{D}}(X \geq \theta + C\sigma\epsilon) = \mathbb{P}(Z \geq C\epsilon) = 1 - \Phi(C\epsilon),$$

where $\Phi$ is the CDF of $Z$. We want to show that $1 - \Phi(C\epsilon) < 1/2 - \epsilon$ for all $\epsilon < 2/7$, which is equivalent to $\Phi(C\epsilon) > 1/2 + \epsilon$. On the other hand, we have

$$\Phi(C\epsilon) - 1/2 = \Phi\left(C\epsilon\right) - \Phi\left(0\right)$$
$$= \frac{1}{\sqrt{2\pi}} \int_0^{C\epsilon} e^{-z^2/2} dz$$
$$\geq \frac{1}{\sqrt{2\pi}} \int_0^{C\epsilon} 1 - z^2/2 \, dz$$
$$= \frac{1}{\sqrt{2\pi}} \left( C\epsilon - \frac{C^3\epsilon^3}{6} \right).$$

Hence, proving that $\Phi(C\epsilon) - 1/2 > \epsilon$ is reduced to showing that $C - (C^3\epsilon^2)/6 > \sqrt{2\pi}$, which holds for $C = 4$ and all $\epsilon \in (0, 2/7]$. Since we have that $\mathcal{D}_\theta$ is $(\mathcal{O}(\sigma\epsilon), \epsilon)$-strictly smooth for $\epsilon < 2/7$ we have

$$s\left( \epsilon + \sqrt{\frac{\ln 2/\delta_0}{2n}} \right) \leq \mathcal{O}\left( \left( \epsilon + \sqrt{\frac{\ln 2/\delta_0}{2n}} \right) \sigma \right).$$

For sufficiently large $n$, we can rewrite this inequality as

$$s\left( \epsilon + \sqrt{\frac{\ln 2/\delta_0}{2n}} \right) \leq \mathcal{O}\left( \left( \epsilon + \sqrt{\frac{\ln 1/\delta_0}{n}} \right) \sigma \right).$$

and the desired inequality on $R_{agn}$ follows directly.

In the case $\epsilon = 0$ one can directly use Eq. (5) from the proof of Theorem 4.4 to directly bound the distance between the sample median $m$ and the true mean $\theta$, reaching the same upper bound. Thus, the desired inequality holds for all $\epsilon \in [0, 2/7]$. □

### B.3 Proof of Corollary 5.2

**Corollary 5.2.** *In the context of Theorem 5.1, assume that either of the following two conditions holds:*

    *i) $\mathcal{U}_{red}$ is bounded,*

    *ii) There exists $L > 0$ such that $\mathcal{U}_{red}(x, y) \leq L|x - y|$ for all $x \geq y \geq 0$ and $R_{agn}/n^\alpha \overset{\mathbb{P}}{\to} 0$ as $n \to \infty$.*

*Then, for sufficiently large $n$, Eq. (9) holds with high probability over all randomness.*

*Proof.* We prove both parts independently as follows.

**i)** If there exists some $M > 0$ such that $0 \leq \mathcal{U}_{red}(x, y) < M$ for all $x \geq y \geq 0$, then $P(F) \leq M$ and hence

$$\left[ \frac{|\mathcal{L}|}{C(|\mathcal{L}| - 1)} P(F) \right]^{1/\alpha} \leq \left[ \frac{|\mathcal{L}|M}{C(|\mathcal{L}| - 1)} \right]^{1/\alpha}.$$

Hence, for sufficiently large $n$, the condition of Eq. (9) is satisfied.

**ii)** Suppose that such $L$ exists and $R_{agn}/n^\alpha \xrightarrow{\mathbb{P}} 0$ as $n \to \infty$. Then

$$
\begin{aligned}
P(F) &:= \frac{1}{|\mathcal{L}|} \sum_{l \in \mathcal{L}} \mathcal{U}_{red}\left(R_l(F), R_l(F_l^*)\right) \\
&\leq \frac{L}{|\mathcal{L}|} \sum_{l \in \mathcal{L}} |R_l(F) - R_l(F_l^*)| \\
&\leq \frac{L}{|\mathcal{L}|} \sum_{l \in \mathcal{L}} R_l(F) \\
&\leq L R_{agn}(F).
\end{aligned}
$$

The second line follows from the existence of such $L$, the third line follows from $R_l(F) \geq R_l(F_l^*) \geq 0$ and the last line follows from the definition of $R_{agn}$. Furthermore, we have $R_{agn}(F)/n^\alpha \xrightarrow{\mathbb{P}} 0$, hence for sufficiently large $n$, we have

$$
\frac{P(F)}{n^\alpha} \leq \frac{L R_{agn}(F)}{n^\alpha} \leq \frac{C(|\mathcal{L}| - 1)}{|\mathcal{L}|}
$$

with high probability. Rewriting this, we get

$$
\left[ \frac{|\mathcal{L}|}{C(|\mathcal{L}| - 1)} P(F) \right]^{1/\alpha} < n
$$

with high probability, as desired. $\qquad\square$

## C  Further experimental details

Appendix C.1 contains additional details of the experiments presented in Section 5.1.

### C.1  Additional details for real-world data experiments

For the CIFAR-10 dataset, we use a custom CNN that we present in Table 2. All instances of our model are trained using Adam optimizer with learning rate $1e-3$ and batch size 128.

For experiments on CIFAR-10 with label noise, prefiltering is done by training the standard ResNet-9 architecture using batch size 128 and Adam optimizer with learning rate $1e-3$ for 7 epochs. For experiments with shortcuts, the model we use for prefiltering is our custom CNN, trained using batch size 128 and learning rate $1e-3$ for 5 epochs.

For the Adult dataset, we consider seven different algorithms, as described in Section 5.1. All models that are implemented using scikit-learn use the default values provided by the package. The two-layer fully-connected neural networks use batch size 64, learning rate $1e-3$, and are trained for 15 epochs. In Fig. 1 we show the individual performances of all models.

In all experiments we calculate the price of learner-agnostic prefiltering as follows. First, we conduct multiple runs where we run our prefiltering procedure with different values of the prefiltering hyperparameter $p$ on a noisy training and validation set. In each run we train all models on the prefiltered train and validation datasets and evaluate on the clean risk evaluation dataset. Finally, for each run we compute the best learner-specific risk over all $p$, and then we use that to calculate the price of learner-agnostic prefiltering for each of the preset prefiltering procedures. Finally, we report the smallest price of learner-agnostic prefiltering, averaged over the specific number of runs.

When we report the price of learner-agnostic prefiltering as a function of subset size $k$ in Fig. 2, we do so by taking all subsets of size $k$ of the initially fixed learner set and then computing for each run the average price of learner-agnostic prefiltering over all such subsets. Finally we present means and errors over all runs.

Table 2: CNN architecture (width $w = 32$, #classes $= 10$).

| Layer | Operation | Output Shape |
|---|---|---|
| Input | – | $C \times 32 \times 32$ |
| Conv0 | Conv2D($C$, $w$, 3×3, padding=1) | $w \times 32 \times 32$ |
| ReLU0 | ReLU | $w \times 32 \times 32$ |
| Conv1 | Conv2D($w$, $2w$, 3×3, padding=1) | $2w \times 32 \times 32$ |
| ReLU1 | ReLU | $2w \times 32 \times 32$ |
| Conv2 | Conv2D($2w$, $4w$, 3×3, stride=2, padding=1) | $4w \times 16 \times 16$ |
| ReLU2 | ReLU | $4w \times 16 \times 16$ |
| Pool0 | MaxPool(3×3) | $4w \times 14 \times 14$ |
| Conv3 | Conv2D($4w$, $4w$, 3×3, stride=2, padding=1) | $4w \times 7 \times 7$ |
| ReLU3 | ReLU | $4w \times 7 \times 7$ |
| Pool1 | AdaptiveAvgPool(1×1) | $4w \times 1 \times 1$ |
| Flatten | Flatten | $4w$ |
| Linear | Linear($4w$, 10) | 10 |

# D  Additional experiments

This section contains additional experiments and is structured as follows:

- Appendix D.1 contains empirical analysis of several prefiltering procedures in the context of the scalar mean estimation setup.

- Appendix D.2 contains experiments on CIFAR-10 with contamination based on shortcuts

- Appendix D.3 contains experiments on CIFAR-10N.

- Appendix D.4 contains experiments with Confident Learning.

- Appendix D.5 contains experiments with oracle prefiltering procedures.

- Appendix D.6 experiments with Tiny ImageNet.

- Appendix D.7 contains the individual learner performance from the setups presented in Section 5.1.

- Appendix D.8 explores the dependence of $P(F)$ on learner diameter and number of models.

- Appendix D.9 contains experiments on CIFAR-10 with shortcuts with different prefiltering mechanisms.

- Appendix D.10 contains experiments on CIFAR-10 with shortcuts and different learner sets

## D.1  Empirical analysis of the scalar mean estimation setup

In this subsection we empirically explore the dependence of $R_{agn}$ on $\epsilon$ and $\Delta$ with $\mathcal{D}_\theta = \mathcal{N}(0,1)$ (and hence $\theta = 0$), by analyzing three different prefiltering procedures. The sample size we set is $n = 10001$. We simulate Huber contamination by using a set of noise distributions $\mathfrak{Q} = \{\mathcal{N}(m,1) : m \in \mathbb{R}\}$. We can use symmetry arguments to restrict the space to $m \geq 0$, and we empirically observe that considering 50 equidistant values in $[0, 10]$ for $m$ recovers the worst-case contamination in all experiments.

The first prefiltering procedure is $F_p^q$ as defined in Section 4.1.2, the other two are analogously defined using z-score (Maronna et al., 2006) and Stahel-Donoho outlyingness (SDO) (Stahel, 1981; Donoho, 1982; Zuo et al., 2004).

The second prefiltering that we study is based on the famous "three-sigma edit" rule. Intuitively speaking, given a sample $x_1, \ldots, x_n$, we can measure the "outlyingness" of a single observation as $t_i = (x_i - \bar{x})/s$ and then remove all points for which $|t_i| > 3$. This measure is also known as the z-score of the observation. Motivated by this, we study the sample z-score function as

$$Z\left(x, \{X_1, \ldots, X_n\}\right) := \left| \frac{x - M(X_1, \ldots, X_n)}{SD(X_1, \ldots, X_n)} \right|$$

where the sample mean $M(X_1, \ldots, X_n)$ and the sample standard deviation $SD(X_1, \ldots, X_n)$ are defined as

$$M(X_1, \ldots, X_n) = \frac{1}{n} \sum_{k=1}^{n} X_k$$

$$SD(X_1, \ldots, X_n) = \sqrt{\frac{1}{n} \sum_{k=1}^{n} (X_k - M(X_1, \ldots, X_n))^2}.$$

Then, we can define the z-score prefiltering mechanism as

$$F_l^z(S) := \{X \in S : Z(X, S) < l\}.$$

Note that this mechanism has a hyperparameter $l$ that can be selected in order to apply an "$l$-sigma edit" rule on a dataset.

The third prefiltering mechanism is based on Stahel-Donoho outlyingness (SDO), which can be defined as

$$SDO(x, \{X_1, \ldots, X_n\}) := \frac{|x - Med(X_1, \ldots, X_n)|}{MAD(X_1, \ldots, X_n)},$$

where $Med(X_1, \ldots, X_n)$ and $MAD(X_1, \ldots, X_n)$ are defined as the sample median and the median absolute deviation respectively.

$$Med(X_1, \ldots, X_n) = \begin{cases} X_{(k)}, & \text{if } n \text{ is odd, } k = \frac{n+1}{2} \\ \frac{X_{(k)} + X_{(k+1)}}{2}, & \text{if } n \text{ is even, } k = \frac{n}{2} \end{cases}$$

$$MAD(X_1, \ldots, X_n) = Med\left(\{|X_i - Med(X_1, \ldots, X_n)|\}\right).$$

This induces the SDO prefiltering procedure, defined as

$$F_p^{sdo}(S) := \{X \in S : SDO(X, S) < p\}.$$

The hyperparameter $p$ can take values in the range $(0, \infty)$, with similar interpretation as in the "$p$-sigma edit" rule with different measures for location and scale.

Each procedure is being optimized over its own hyperparameter that controls the amount of data being removed.

In Fig. 5a we calculate the learner-agnostic risk for each prefiltering procedure as a function of $\epsilon$ as follows. The learner set is $\mathcal{L} = \{H_\delta : \delta \in \Delta\}$ where $\Delta = \{0.01, 1.0\}$. First, for each value of the prefiltering hyperparameter $p$ and noise parameter $m$ we calculate the worst-case distance for each Huber learner. Then, we take the minimax value by first maximizing over the noise parameter $m$ and then minimizing by the prefiltering hyperparameter $p$. That way we get a guarantee on the learner-agnostic risk. Finally, we repeat this process 8 times in order to get the expected value and standard error of $R_{agn}$.

We notice that the empirical trend exhibits superlinear growth, in alignment with the $\epsilon^2$ term in Theorem 4.4. This is also consistent with lower bounds that we find in (Diakonikolas & Kane, 2023). We also observe that $F^z$ and $F^{sdo}$ outperform $F^q$ in this setup. In any case, our results show that learning is viable under the assumption of reasonable downstream learners.

In Fig. 5b we measure the second term in Eq. (2) on a set $\mathcal{L} = \{\delta_1, \delta_2\}$ where $\epsilon = 0.2$, $\delta_1 = 0.01$ and $\delta_2$ varies along the x-axis, thus creating varying levels of heterogeneity between the learners. A single run of our experiments consists of generating a sample, prefiltering with a specific prefiltering procedure with a specific hyperparameter $p$ and then producing estimates using the downstream learners. Then we record the risk for each learner separately. Finally, we report the respective risk minimized over the prefiltering hyperparameter $p$. We see that increasing learner heterogeneity leads to larger additional loss incurred by the models. Moreover, the behavior of this term is different for the different prefiltering procedures, suggesting that the optimal choice might depend on the learner set. This aligns with the analysis presented in Section 4.1.1, where we argued that the design of procedures for LARP should depend on the downstream learner set. In all cases, we conclude that the effect of the learner set is statistically significant and the guarantees for downstream learners are inherently worse due to learner heterogeneity.

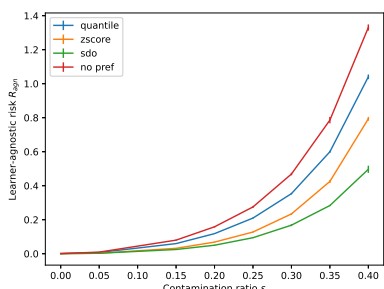 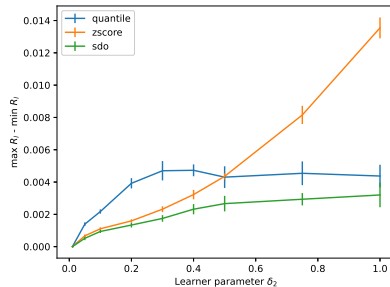

(a) Learner-agnostic risk as a function of (b) Gap between best-case and worst-case
$\epsilon$.                                                                 risk.

Figure 5: Experiments for Gaussian mean estimation.

## D.2 Experiments on real-world data with noise based on shortcuts

In order to study the generality of our results with respect to the noise model, we provide a modification of our setup in which the noise is based on shortcuts in the data. This simulates scenarios where training data contains spurious correlation features irrelevant to the labels, e.g., textual information embedded in image patches (Geirhos et al., 2020; Shah et al., 2020; Nam et al., 2020; Sagawa et al., 2020b).

For tabular data, we inject noise by maximizing the integer feature "education-num" in 85% of the data for the class corresponding to the label ">50K", leading to $\epsilon \approx 21\%$ contamination rate due to class imbalance. For image data, we set the color of a $3 \times 3$ patch in the lower right corner of 85% of the data in classes 0, 2, 4 according to their label, resulting in $\epsilon = 25.5\%$ contamination rate.

Based on the observation that models learn shortcut data quickly, in both cases we prefilter the top-$p$ percent of lowest-loss data points based on CNN (CIFAR-10) and Random Forests (Adult) models from the respective learner sets, but trained on the combined train and validation sets. We verify that this results in $> 85\%$ of prefiltered shortcut data in practice at the default $p = 25\%$. We use the macro-F1 score (Manning et al., 2008) as our risk metric $R$ due to the class imbalance of the prefiltered datasets.

For experiments on CIFAR-10 we provide a different notion of learner heterogeneity. Each learner regularizes the effect of the shortcuts on the training explicitly by reducing the gradient contribution of the shortcut patch pixels by a factor in the range $[1e-6, 1]$.

In all shortcut experiments, learners reweight their losses to account for class imbalance of the prefiltered sets.

Results are presented in Fig. 6. We see that the results are aligned with the observed tendencies in Section 5.1, supporting the generality of our conclusions with respect to the contamination model. In Appendix D.8, Appendix D.9, and Appendix D.10 we present additional studies on this setup, including different prefiltering procedures, different learning sets, and ablation studies on the effect of learner heterogeneity on the price of LARP.

## D.3 Experiments on CIFAR-10N

In this subsection we provide a modification of our main experiments in Section 5.1 on image data by substituting the synthetic noise model that applies uniform label noise, with labels which contain real-world annotation errors. We do so by using the CIFAR-10N (Wei et al., 2022) dataset suite, which contains human-annotated real-world noisy labels collected from Amazon Mechanical Turk. In particular, we use the "Random 1" version of the label set, which uses the first submitted label for each image. This dataset has a contamination rate of 17.23%, though the distribution of the noise is more complex than a simple independent sample from a uniform distribution. Nevertheless, we observe similar results in our experiments as in the main setup. Results are presented in Fig. 7. We observe similar behavior to the setup with the

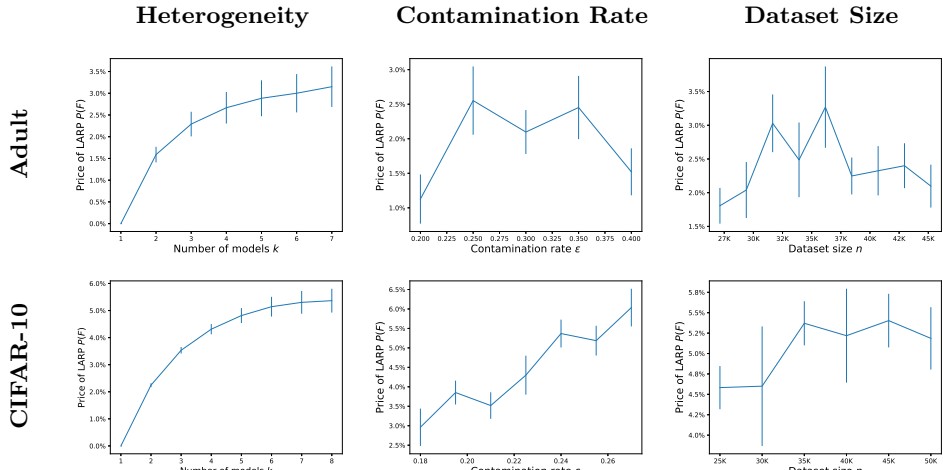

Figure 6: Price of learner-agnostic prefiltering for Adult (upper row) and CIFAR-10 (lower row) in the presence of shortcuts. From left to right: heterogeneity, contamination rate, and dataset size.

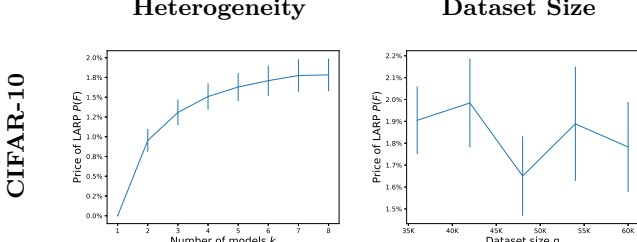

Figure 7: Price of learner-agnostic prefiltering for the CIFAR-10N dataset. From left to right: heterogeneity and dataset size.

synthetic label noise, which confirms our design choice as a useful representation of a more realistic setup where noise is generated from human annotations.

### D.4 Experiments with Confident Learning

In this subsection we provide a modification of our setup, where prefiltering is based on the Confident Learning (CL) approach. First, we note that CL is a method for binary classification in the presence of label noise. In particular, CL ranks training points by how likely their observed labels are corrupted using two ingredients: calibrated out-of-sample class-probability estimates and class-conditional "confident" thresholds. First, we fit a probabilistic classifier to the noisy labels and obtain out-of-sample predictions. Then we rank points by their inconsistency with their observed class and then we remove the top-ranked (most outlying) data points. We extend this setup for the multiclass setting on CIFAR-10 as follows. First we train a ResNet-9 model, modeling the noisy data distribution. The trained model is then used to predict class probabilities for every sample in the training set, defining its confidence score. Finally, the top $p\%$ with the highest confidence scores are removed. Results are presented in Fig. 8.

Once again, we see that our results are consistent with our main setup, further indicating the generality of our results. We note that $P(F)$ is decreasing as we increase $\epsilon$, which is also visible in the larger $\epsilon$ rates in other experiments. This further solidifies that the relationship between $\epsilon$ and the price of LARP is complex and requires further exploration.

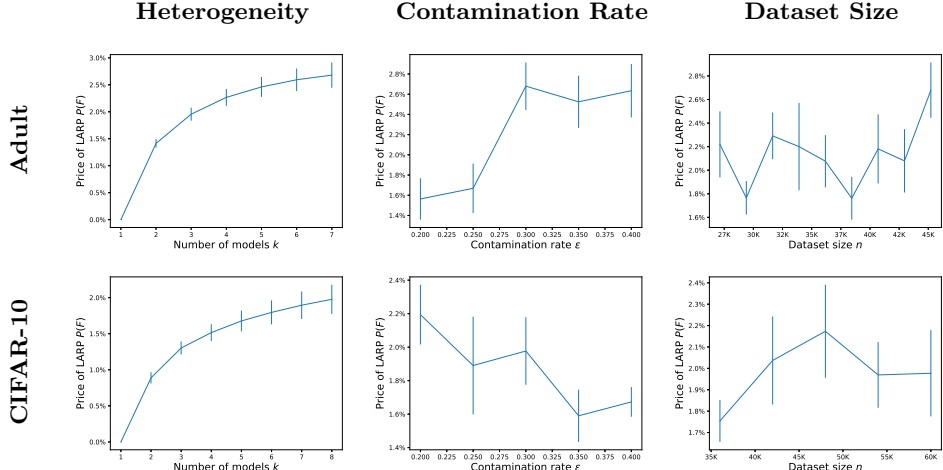

Figure 8: Price of learner-agnostic prefiltering for Adult (upper row) and CIFAR-10 (lower row) in the presence of label noise for prefiltering procedure based on Confident Learning. From left to right: heterogeneity, contamination rate, and dataset size.

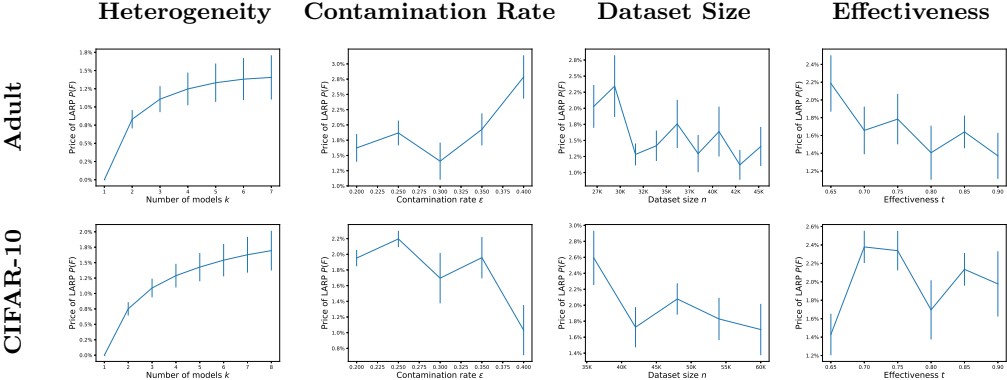

Figure 9: Price of learner-agnostic prefiltering for oracle prefiltering procedures. From left to right: heterogeneity, contamination rate, dataset size, and effectiveness.

### D.5 Experiments with oracle prefiltering procedures

In this subsection we measure the price of LARP for two instances of the oracle prefiltering procedure $F_p^{o,t}$, one for the CIFAR-10 and Adult datasets each. Both datasets are contaminated with uniform label noise as in the canonical setting. We implement these oracle prefiltering procedures by simply recalculating the sample size and the contamination rate after conducting the prefiltering procedure, which is equivalent to applying the filter as described in Section 4.2. Results are presented in Fig. 9. We see that the trends of the price of LARP with respect to the learner heterogeneity, the contamination rate and the dataset size are similar to what we observe with other, more practical prefiltering procedures. Furthermore, we also plot the price of LARP $P(F)$ against the effectiveness $t$ of the prefiltering. Most interestingly, we see that the price of LARP is highest when an oracle prefiltering procedure is neither too weak nor too strong for the task. Intuitively, this is reasonable as we expect that for small $t$ all learners prefer no prefiltering at all ($p \approx 0$), which leads to a smaller price of LARP.

### D.6 Experiments with Tiny ImageNet

In order to show our methods are applicable to larger datasets we provide a new experiment in the same setting as our CIFAR-10 setup with label noise, but the training dataset is swapped with Tiny ImageNet.

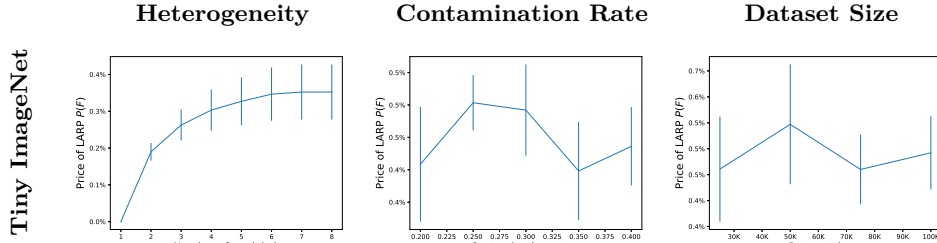

Figure 10: Price of learner-agnostic prefiltering for the Tiny ImageNet dataset. From left to right: heterogeneity, contamination rate, and dataset size.

The Tiny ImageNet dataset uses 100K images across 200 classes from the original ImageNet, reduced to 64x64 resolution. The learner set consists of the same CNN model, but its width is set to 64 and the range of the L2 regularization is reduced to [1e-6,1.5e-3] due to the increased complexity of the dataset. The model in the prefiltering procedure is also modified according to the increased image width and number of classes.

Results are presented in Fig. 10. Once again, we see that our results align with our main findings. However, we also observe a weaker signal in all cases. We conjecture that this is due to the fact that the task is inherently harder in the presence of such a large number of label classes. This makes all downstream performance worse, and hence we presume that this diminishes the difference between generalization performance of models after different amounts of data are being prefiltered.

### D.7  Learner-specific risks

In this subsection we present the ability of the prefiltering procedure to protect all learners individually. On each plot, we will present the risk evaluation metric of each model in the absence of noise ($\epsilon = 0$), in the presence of noise but no prefiltering ($\epsilon > 0$, no pref), and in the presence of noise and optimal learner-specific prefiltering ($\epsilon > 0$, spec). The results for the Adult dataset are presented as a histogram, where on the x-axis we have all seven models, each described by their algorithm.

We also plot the same metrics for the tasks on the CIFAR-10 dataset. This time, we plot all values of the regularization hyperparameter on the x-axis, and then we plot the same metrics for each learner. Results are shown in Fig. 1.

### D.8  Dependence of $P(F)$ on learner diameter and number of models

We consider further parametric studies of the impact of learner heterogeneity on the price of learner-agnostic prefiltering. Let us parametrize each learner by the $\log_{10}$ value of its regularization parameter. We conduct the same setups as CIFAR-10 + label noise and CIFAR-10 + shortcuts, but we now consider 25 learners whose learner parameters $c_1, \ldots, c_{25}$ are equidistant in an interval ($[-3.42, -2.26]$ for the case of label noise, and $[-4.43, -1]$ for the case of shortcuts). Then, for parameters $m$ and $d$, we fix 25 learner sets, each centered at one of the aforementioned learners, such set $i$ is centered at $c_i$, contains $m$ models whose parameters are equidistant in $[c_i - cd(m-1)/2, c_i + cd(m-1)/2]$, where $c = c_2 - c_1 = 0.14$. Then we compute the average price of learner-agnostic prefiltering over all of the 25 learner sets. By only varying one of $m$ and $d$ we can isolate the dependence on the diameter of the learners, either in terms of the parameters or the number of learner sets. Results are presented in Fig. 11 for the setting of label noise and Fig. 12 for the setting of shortcuts. We see that both the size and diameter of the learner set contribute to the increase in price of learner-agnostic prefiltering, but the contribution of the size of the learner set is more significant.

### D.9  Experiments on CIFAR-10 with shortcuts with different prefiltering mechanisms

For the setting of CIFAR-10 with shortcuts, we also conducted a single experiment in the canonical setting of $\epsilon = 25.5\%$, with a different prefiltering procedure. The new prefiltering procedure trains an instance of our custom CNN with early stopping. During training, the prefiltering procedure tracks the number of epochs

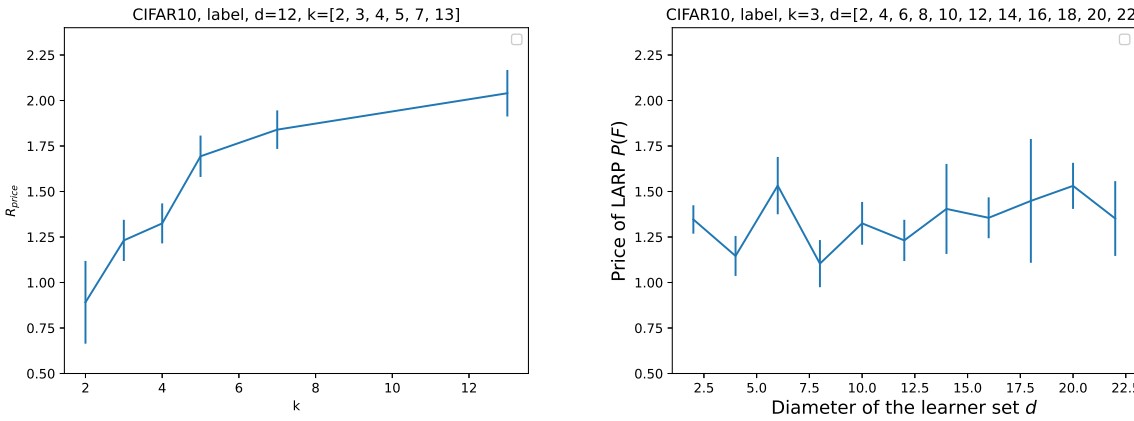

Figure 11: Price of learner-agnostic prefiltering for CIFAR-10 with label noise.

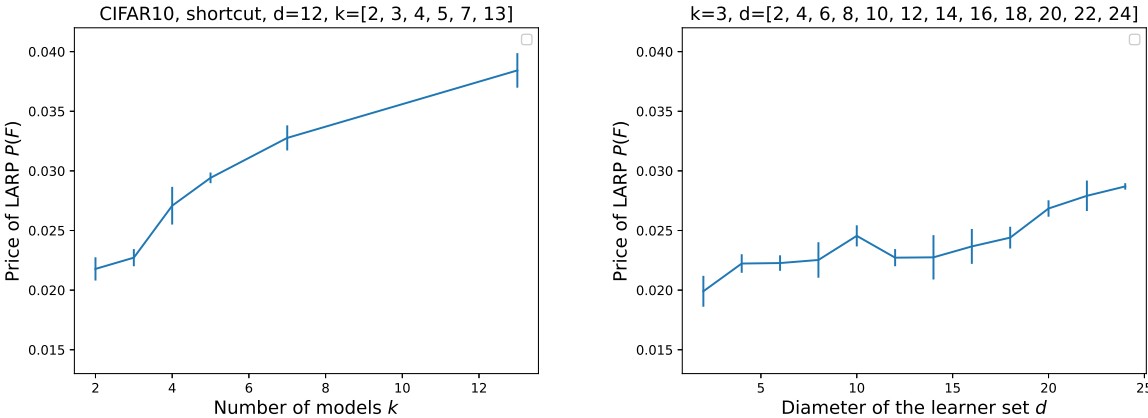

Figure 12: Price of learner-agnostic prefiltering for CIFAR-10 with shortcuts.

to reach loss value $< 0.01$ for each data point in the training set, and then the top-$p$ fraction of the points with the lowest number of epochs to low loss value is removed. This again works under the assumption that models fit more quickly spurious correlations that stem from the shortcut patch. Results are presented in Fig. 13. We also present a variation where we use a two-layer neural network instead of our CNN. We present the results in Fig. 14.

We see that the two variations of the prefiltering procedure are also effective at protecting each learner individually. Nevertheless, we observe a significant price of learner-agnostic prefiltering. We use this experiment as an indication that the significance of the price of learner-agnostic prefiltering is present across different prefiltering procedures.

These plots suggest that our prefiltering mechanisms can be useful for protecting individual downstream learners. By reasoning about the usefulness in the learner-specific regime, combined with Theorem 5.1, we can argue about the general benefit of conducting learner-agnostic prefiltering in general. We consider the derivation of provable guarantees on prefiltering an exciting idea for future work.

## D.10 Experiments on CIFAR-10 with shortcuts and different learner sets

In this subsection we present another variation of the setup presented in Appendix D.2. We consider a different learner set that is again based on our custom CNN. Recall that in Appendix D.2, all learners reweight their

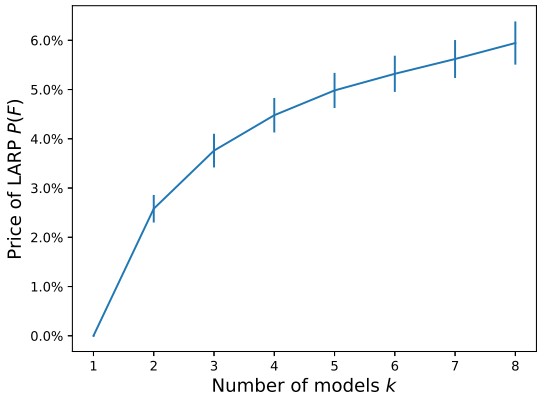 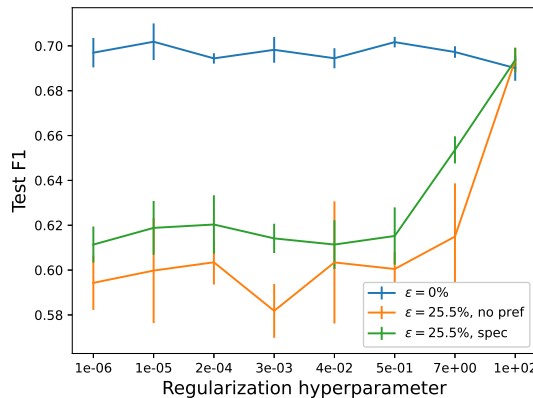

(a) Price of learner-agnostic prefiltering as a function of learner heterogeneity.

(b) F1 Accuracy in the presence of no noise (blue), $\epsilon = 25.5\%$ (orange), and $\epsilon = 25.5\%$ with learner-specific prefiltering.

Figure 13: An instance of LARP with prefiltering procedure that computes time to low loss for our custom CNN.

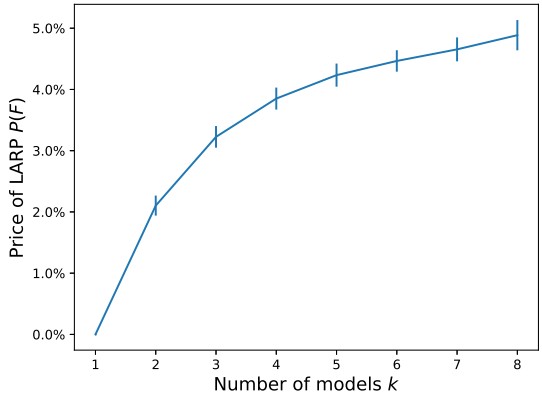 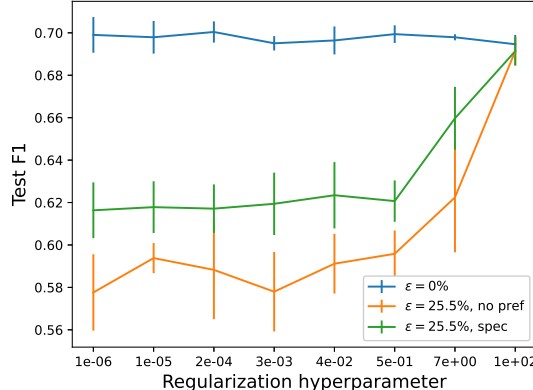

(a) Price of learner-agnostic prefiltering as a function of learner heterogeneity.

(b) F1 Accuracy in the presence of no noise (blue), $\epsilon = 25.5\%$ (orange), and $\epsilon = 25.5\%$ with learner-specific prefiltering.

Figure 14: An instance of LARP with prefiltering procedure that computes time to low loss for a two-layer neural network.

losses during training proportionally to the class distribution $c_1, \ldots, c_{10}$. Now, all models are parametrized by a scalar $w$ which dictates how to reweight training losses in accordance with the class imbalance present in the prefiltering set. In particular, we consider 10 models whose values for $w$ are equidistant in the range $[0, 2]$, and each $w$ corresponds to a model that reweights losses proportionally to $c_1^w, \ldots, c_{10}^w$ where $c_1, \ldots, c_{10}$ represent the proportions of each class present in the dataset. For example, when $w = 0$, the model does not do any loss reweighting. The models in Appendix D.2 correspond to $w = 1$, and $w = 2$ reweights aggressively to be even more considerate of minority classes. We empirically observe that for small values of $w$, which correspond to models with no loss reweighting, the class imbalance of a perfectly prefiltered dataset is more detrimental than the generalization loss from the spurious correlation. On the other hand, large values of $w$ can handle class imbalance well, leading to a preference for more aggressive prefiltering. This preference

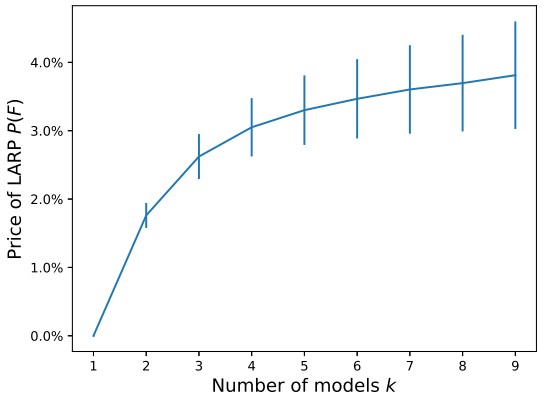 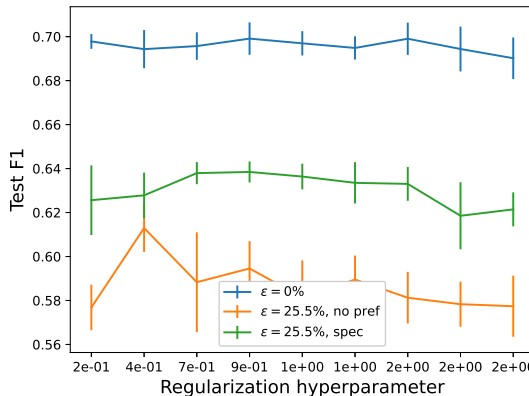

(a) Price of learner-agnostic prefiltering as a function of learner heterogeneity.

(b) F1 Accuracy in the presence of no noise (blue), $\epsilon = 25.5\%$ (orange), and $\epsilon = 25.5\%$ with learner-specific prefiltering.

Figure 15: An instance of LARP with a learner set defined using loss reweighting coefficients.

heterogeneity induces the price of LARP. We present the price of learner-agnostic prefiltering as well as learner-specific risks in Fig. 15.

## E    Further discussion on Algorithm 1

In this section we discuss how Algorithm 1 captures our empirical setups, as well as real-world data cleaning procedures.

Our general description of Algorithm 1 matches the general structure of the prefiltering procedures used for the experiments with label noise (Section 5.1, Appendix D.4, Appendix D.6), human label noise (Appendix D.3), and shortcuts (Appendix D.2, Appendix D.9, Appendix D.10). In particular, we have the following realizations:

- In label noise experiments on CIFAR-10, $\mathcal{M}$ is ResNet-9, and $g(z, \mathcal{M}_\theta)$ is the loss of $\mathcal{M}_\theta$ on $z$.

- In label noise experiments on Adult, $\mathcal{M}$ is two-layer NN, and $g(z, \mathcal{M}_\theta)$ is the loss of $\mathcal{M}_\theta$ on $z$.

- In shortcut experiments on CIFAR-10, $\mathcal{M}$ is our custom CNN, and $g(z, \mathcal{M}_\theta)$ is the negative of the loss of $\mathcal{M}_\theta$ on $z$.

- In shortcut experiments on Adult, $\mathcal{M}$ is Random Forest, and $g(z, \mathcal{M}_\theta)$ is the negative of the loss of $\mathcal{M}_\theta$ on $z$.

- In label noise experiments on Tiny ImageNet (Appendix D.6), $\mathcal{M}$ is ResNet-9, and $g(z, \mathcal{M}_\theta)$ is the loss of $\mathcal{M}_\theta$ on $z$.

- In label noise experiments using Confident Learning (Appendix D.4), $\mathcal{M}$ is ResNet-9, and $g$ is the confidence score adapted from Northcutt et al. (2021a).

- In additional experiments on CIFAR-10 with shortcuts (Appendix D.10), we use two variations of the implementation presented in Appendix D.2. In the first one we change $\mathcal{M}$ to be a two-layer NN, keeping the same $g$; in the second one we use the original $\mathcal{M}$, but the scoring function $g$ tracks the number of epochs until a specific point reaches training loss below 0.01.

This algorithm also covers the prefiltering procedure presented in Section 5.2, up to some modifications. Since we are concerned with doubly imbalanced datasets, our prefiltering procedure reweights the data in order to

---

**Algorithm 2** Prefiltering Procedure for Doubly Imbalanced Datasets

---

**Require:** Provider Dataset $S$, hyperparameters $\alpha, \beta, \gamma$, scoring function $g(\cdot, \cdot)$, and scoring model/estimator $\mathcal{M}$

1: $G_1, ..., G_4 \leftarrow Split(S)$          // Split in subgroups
2: $p_1, ..., p_4 \leftarrow SubgroupQuantiles(\alpha, \beta, \gamma)$          // Subsampling proportions
3: **for** $i \in 1, ..., 4$ **do**
4:      $G'_i \leftarrow SubSample(G_i, p_i)$          // Subsample each subgroup
5: **end for**
6: $S' \leftarrow G'_1 \cup G'_2 \cup G'_3 \cup G'_4$          // Combine prefiltered subgroups
7: **return** $S'$

---

improve fairness (Yalcin et al., 2025). This approach splits the dataset into 4 subgroups according to the two axes of imbalance: label and sensitive attribute. Thus, the effective parametrization of the algorithm contains 3 degrees of freedom ($\alpha, \beta, \gamma$ from Yalcin et al. (2025)), which in turn set 4 quantile thresholds, one for each subgroup. This is in contrast to Algorithm 1, where only one quantile hyperparameter $p$ is given as input. We provide a specialized pseudocode of this prefiltering procedure in Algorithm 2.

We also note that Algorithm 1 captures the high-level structure of the prefiltering procedures used for mean estimation (Section 4.1, Appendix D.1). It also covers the oracle prefiltering procedures (Section 4.2, Appendix D.5) under the idealistic assumption that the prefiltering procedure, through the scoring function $g$, also has access to whether each data point is contaminated or not.

Finally, Algorithm 1 bears resemblance to standard data filtering pipelines used to curate modern large-scale datasets. For example, LAION-5B (Schuhmann et al., 2022) uses a pre-trained CLIP model ($\mathcal{M}$) (Radford et al., 2021) to filter image-text pairs. Similarly, Raffel et al. (2020) employ a language identification model ($\mathcal{M}$) to curate the Common Crawl in the process of creating the C4 dataset. Therefore, varying the definitions of $\mathcal{M}$ and $g$ in Algorithm 1 can describe common heuristic prefiltering practices of foundation-model datasets, similarly to the loss-based filtering used in our experimental analysis. Note that setting specific thresholds on $g$ is effectively setting what fraction $p$ of outliers with the highest scores should be filtered out.

## F   Discussion on the sufficient conditions in Corollary 5.2

In this section we discuss the utility of Corollary 5.2 through its connection to realistic examples as well as the upper bounds derived in Theorem 4.4 and Theorem 4.6. This serves as further indication of the overall benefit of learner-agnostic prefiltering.

First, condition i) of Corollary 5.2 is satisfied in cases where the utility of our model is bounded. One classic example is a model in which utility is directly proportional to the generalization accuracy of the model. One real-world use-case for such utility is a recommendation model in which fixed utility is gained for each successful recommendation.

On the other hand, we can show a strong connection between condition ii) in Corollary 5.2 and the upper bounds provided in Section 4.

One way to argue about such benefit is by considering the in-probability bounds in Section 4. Recall that in Corollary 5.2 we show that as long as the learner-agnostic risk converges to 0 in probability and the utility function is Lipschitz, the condition of Theorem 5.1 is satisfied for sufficiently large $n$. This is relevant to both theoretical setups provided in Section 4 since we provide upper bounds with high probability in Eq. (4) and Eq. (8). Note that in both cases the right-hand sides, which we informally denote as $RHS_{mean}$ and $RHS_{oracle}$, satisfy $RHS_{mean}/n^\alpha \to 0$ and $RHS_{oracle}/n^\alpha \to 0$. Hence, we can use those bounds to show that $R_{agn}(F)/n^\alpha \xrightarrow{\mathbb{P}} 0$ as $n \to \infty$.

We can also provide an alternative argument as follows. The choice of whether a learner uses learner-specific or learner-agnostic prefiltering should realistically happen before any learning is done. Therefore, it is not practical to assume that the risks themselves will be known at this point in time. Instead, we can use

the upper bounds as a surrogate measure of downstream performance. In this case, we can see that the convergence as $n \to \infty$ is direct and the same conclusion follows.

Hence, in both setups we can deduce that learner-agnostic prefiltering is beneficial for sufficiently large datasets so long as the utility reduction function $U_{red}$ satisfies the inequality presented in condition ii). One classic example of that is when we define this function as

$$U_{red}(x, y) = \mathcal{R}(y) - \mathcal{R}(x)$$

for all $x \geq y \geq 0$, where $\mathcal{R} : \mathbb{R}_{\geq 0} \to \mathbb{R}$ is a Lipschitz-continuous utility function. Informally, $\mathcal{R}$ measures the utility of a model as a function of its downstream risk, i.e., $\mathcal{R}$ is decreasing in its argument. This captures a rich family of utility reduction functions which can be combined with our theoretical results to provide guarantees for the benefit of LARP for sufficiently large sample sizes.

## G   Miscellaneous

This section contains miscellaneous information and is structured as follows:

- Appendix G.1 contains the computational requirements for the experiments.

- Appendix G.2 contains descriptions of licenses of the datasets used in the paper.

### G.1   Compute resources

Conducting the experiments required around 250 GPU hours on 16 NVIDIA L4 GPUs with 24GB VRAM each for experiments on CIFAR-10, as well as around 100 CPU hours on a 64-core CPU node for experiments on mean estimation and Adult.

### G.2   Licenses

**CIFAR-10**   CIFAR-10 (Krizhevsky, 2009) is distributed by the University of Toronto; we accessed it via `torchvision`'s CIFAR10 loader. The original distributor provides no explicit license, so we use the dataset for non-commercial research purposes only. Source: `https://www.cs.toronto.edu/~kriz/cifar.html`

**Adult Dataset (Census Income)**   The Adult dataset (Becker & Kohavi, 1996) is derived from the 1994 U.S. Census and is made available through the UCI Machine Learning Repository under the Creative Commons Attribution 4.0 International (CC BY 4.0) license. Source: `https://archive.ics.uci.edu/dataset/2/adult`

**CIFAR-10N**   The released dataset CIFAR-10N (Wei et al., 2022) is publicly available under the Attribution-NonCommercial 4.0 International (CC BY-NC 4.0) license. Source: `http://noisylabels.com`

**Tiny ImageNet**   The Tiny ImageNet dataset is a reduced-version subset of the ILSVRC/ImageNet dataset (200 object classes, images resized to 64x64). The underlying images are subject to the ImageNet terms of access, which grant usage only for non-commercial, research and educational purposes, and require compliance with copyright of the original image owners. While many implementations or loaders of Tiny ImageNet are released under permissive licenses (e.g., MIT) for their code, there is no publicly available license identifying Tiny ImageNet.

**Bank Account Fraud**   The BAF dataset suite (Jesus et al., 2022) is publicly available via Kaggle under the Creative Commons Attribution-NonCommercial-ShareAlike 4.0 International (CC BY-NC-SA 4.0) license. Source: `https://www.kaggle.com/datasets/sgpjesus/bank-account-fraud-dataset-neurips-2022`

