# OpenReview forum: "LARP: Learner-Agnostic Robust Data Prefiltering"
_TMLR — Accepted by TMLR_

### Review · Reviewer_wYKL · 2026-01-12

**Summary Of Contributions:**

The paper considers "learner-agnostic robust data prefiltering" (LARP) that removes "contaminated" samples from a dataset given a set of learners. The paper primarily focuses on the cost of using LARP in place of learner-specific prefiltering that might result in lower risk for only the considered learner.

**Audience:**

Yes

**Audience Explanation:**

Yes, I believe this work is of interest to the statistical machine learning audience.

**Broader Impact Concerns:**

No concerns.

**Claims And Evidence:**

No

**Claims Explanation:**

Frankly, I found this paper hard to read. The claim itself was not clear. I understood the overall goal of removing contaminated samples using a collection of learners such that the resulting filtered dataset has an overall lower risk over the set of considered learners. But I did not understand how exactly this LARP works (both its theoretical and practical design), and the implications of the provided results and theorems. Please see my questions under "Requested Changes."

**Requested Changes:**

High-level comment: I feel that the larger finding of this paper, that a prefiltering technique that works for a set of learners gives worse risk for certain learners compared to a prefiltering technique tailored to that specific learner, is intuitive. In distributional robust optimization (DRO), this trade-off is well-known. But in DRO, robustness over distributions is desirable compared to the utility of individual predictors. Is there such a goal in data prefiltering?

Below are more like questions than requested changes:

Q1. "Learner-agnostic" is a bit of a misnomer, as the prefiltering technique works with a given set of learners. This point is also acknowledged in the paper (sentence starting with "Thus, the optimal procedure may..." on page 4). While the resulting algorithm is agnostic w.r.t. the given set of learners, the term is more suited for a prefiltering technique that does not look into downstream learners at all.
Q2. Why should the prefiltering procedure satisfy $F(S)\subseteq S$ in page 3?
Q3. The theory is on mean estimation, and the experiments are prediction tasks (i.e., conditional mean estimation). Can the authors provide a toy experiment that matches the theory?
Q4. Lemma 4.1 is proven for only the Bernoulli distribution and for Huber estimators that take $\delta$ from a finite set. And this lemma is not used anywhere to prove any theorem.
Q5. What exactly is $s$ in Def. 4.2 and Thm 4.4? It was written $s = s(\epsilon)$.
Q6. What does the quantile prefiltering procedure $Q$ actually do? How do you build it in practice?
Q7. Which prefiltering procedures were used in Sec. 5.1? The text on page 9 stated "all prefiltering procedures," but it did not specify which ones were considered. Please let me know if I missed it.
Q8. I understand that the game theoretic premise in Sec. 5.3 is hypothetical. But why would learner-specific prefiltering be costlier than learner-agnostic prefiltering? In LARP, each learner still has to inform their risk on each sample according to Eq. (1), don't they?

---

> ### Author Response · Authors · 2026-03-13
> **Response to Reviewer wYKL**
>
> We thank the reviewer for their engaging comments and suggestions.
>
> **Q1. Clarifying the paper contribution**
>
> We refer to “Purpose of our work” and “Benefits of LARP” in the main response for a precise description of the goals and scope of our work. We remain available in case of any remaining questions.
>
> **Q2. Learner-agnostic terminology**
>
> We thank the Reviewer for this terminological observation. The term 'agnostic' is intended to reflect that the prefiltering is not tailored to any single learner, but rather suits a broad specification of learners. Our inspiration stems from the software engineering community, where the term “agnostic” implies the ability to work across many settings, though implicitly restricted to some reasonable specifications. For instance, a “platform-agnostic” application may assume a computer with a modern OS, rather than being compatible with any conceivable device. Similarly, in our case, we regard the set of learners as plausible learning algorithms that can be used on top of the dataset. We also argue that working without regard to the set of downstream learners leads to a constant lower bound on the learner-agnostic risk (Lemma 4.1).
>
> **Q3. Why should the prefiltering procedure satisfy $F(S)\subseteq S$ in page 3?**
>
> We use the condition $F(S) \subseteq S$ to formalize data prefiltering, i.e., removing some datapoints out of the initial (possibly corrupted) dataset. By focusing on data prefiltering, we seek to capture common practices of removing outliers and low-quality data, as commonly done in standard curation procedures (e.g., see LAION-5B (Schuhmann et al., 2022) or C4 (Raffel et al., 2020)). Expanding our theoretical framework to additional preprocessing, e.g., by allowing data transformations, is an exciting idea for future work, but beyond the scope of this manuscript, as discussed in Q4 of our response to Reviewer asSb.
>
> **Q4. The theory is on mean estimation, and the experiments are prediction tasks (i.e., conditional mean estimation). Can the authors provide a toy experiment that matches the theory?**
>
> We provide an empirical analysis of the scalar mean estimation setup in Appendix D.1. There, we apply the quantile-based prefiltering from Section 4.1.2, as well as two more prefiltering procedures based on z-scores (Maronna et al., 2006) and the Stahel-Donoho outlyingness measure (Stahel, 1981; Donoho, 1982; Zuo et al., 2004). Then, we compare $R_{agn}$ for the three prefiltering procedures against a baseline where no prefiltering is present. We see that all three prefiltering procedures improve the learner-agnostic risk. We further observe that the value of the gap between best-case and worst-case learner risk depends on the prefiltering procedure.
>
> **Q5. Lemma 4.1 is proven for only the Bernoulli distribution and for Huber estimators that take $\delta$ from a finite set. And this lemma is not used anywhere to prove any theorem.**
>
> We thank the Reviewer for pointing this out. We believe that it is suitable to rename the result to a “proposition”, as it is indeed not used to prove other results. Instead, it is a hardness result that motivates the need to study upper bounds that depend on the learner set.
>
> **Q6. What exactly is $s$ in Def. 4.2 and Theorem 4.4? It was written as $s=s(\epsilon)$.**
>
> Essentially following the notion of smoothness presented in Diakonikolas et al. (2023), $s$ is a function that describes the “sensitivity” of the quantiles that are sufficiently close to the median. This function is a property of the original distribution itself, allowing us to tie the guarantees of Theorem 4.4 to the properties of the clean distribution. For example, as shown in Corollary 4.5, we can see that $\mathcal{N}(\theta, \sigma^2)$ is $(\mathcal{O}(\sigma \epsilon), \epsilon)$-strictly smooth for $\epsilon < 2 / 7$, which allows us to remove the dependence of $s$ in the upper bound on $R_{agn}$. More generally, we can create refined versions of Equation (4) for different distributions $\mathcal{D}$ according to their smoothness.
>
> **Q7. What does the quantile prefiltering procedure $Q$ actually do? How do you build it in practice?**
>
> In the context of Subsection 4.1.2, the quantile prefiltering procedure $F_{p}^{q}$ is a classic method for outlier removal in the context of mean estimation. It works by filtering out the sample points that are outside a specific quantile threshold, controlled by the hyperparameter $p \in (0, 1/2 - 1 / 2n)$. For example, when $p=0.05$, then the quantile prefiltering procedure removes the $5\%$ of the smallest and $5\%$ of the largest samples in the dataset.
> In our experiments, we implement a similar concept (now formalized as Algorithm 1 in the general response) by calculating a score on a surrogate model, and then removing $p\%$ of the most extreme outliers.

---

> > ### Author Response · Authors · 2026-03-13
> > **Response to Reviewer wYKL (Continued)**
> >
> > **Q8. Which prefiltering procedures were used in Sec. 5.1? The text on page 9 stated "all prefiltering procedures," but it did not specify which ones were considered. Please let me know if I missed it.**
> >
> > We would like to thank the Reviewer for pointing this out. The sentence "All prefiltering procedures ..." refers to the prefiltering procedures described in the later parts of the paragraph. We rephrase this in the revised manuscript. We also refer to “Further details on the prefiltering used in the experimental evaluation” for further clarifications on the experiments that we have added in the updated manuscript.
> >
> > **References:**
> >
> > Diakonikolas, Ilias, and Daniel M. Kane. “Algorithmic high-dimensional robust statistics.” Cambridge University Press, 2023.
> >
> > Donoho, David L. “Breakdown properties of multivariate location estimators.” Technical report, Harvard University, Boston, 1982.
> >
> > Maronna, Ricardo A., R. Douglas Martin, and Víctor J. Yohai. "Robust Statistics." Wiley Series in Probability and Statistics (2006): 436.
> >
> > Raffel, Colin, et al. "Exploring the limits of transfer learning with a unified text-to-text transformer." Journal of machine learning research 21.140 (2020): 1-67.
> >
> > Schuhmann, Christoph, et al. "Laion-5b: An open large-scale dataset for training next generation image-text models." Advances in neural information processing systems 35 (2022): 25278-25294.
> >
> > Stahel, Werner A. “Robust Estimation: Infinitesimal Optimality and Covariance Matrix Estimators.” Ph.D. thesis, ETH, Zurich, 1981.
> >
> > Zuo, Yijun, Hengjian Cui, and Xuming He. "On the Stahel-Donoho estimator and depth-weighted means of multivariate data." The Annals of Statistics 32.1 (2004): 167-188.

---

> > > ### Comment · Reviewer_wYKL · 2026-03-31
> > > **Thank you to the authors for their response**
> > >
> > > The authors have adequately addressed my concerns, and their responses are also included in the manuscript.

---

### Review · Reviewer_asSb · 2026-02-18

**Summary Of Contributions:**

This paper study the problem of date prefiltering in order to remove noisy data before training. The paper propose a method called LARP, which minimizes the worst-case loss across a pre-specified set of learners (rather than optimizing for any single learner). Two analytical settings are developed: scalar mean estimation  and a PAC-style oracle setup, providing upper bounds on learner-agnostic risk and a hardness result showing that guarantees must depend on the learner set. The authors further compare learner agnostic vs learner specific data prefiltering. Lastly, the authors validate their theoretical findings on real world datasets including CIFAR, Adult.

**Audience:**

Yes

**Audience Explanation:**

Data selection is a very important area. Formalizing within a theoretically rigorous framework is definitely interested to a group of TMLR's audience.

**Claims And Evidence:**

Yes

**Claims Explanation:**

The paper provides rigorous theoretical analysis, with experiments validating the theoretical bounds.

**Requested Changes:**

- Having a concrete algorithm describing the data prefiltering process, which is applied to the experiments will be helpful for audience to understand.
- The actual classification accuracy for CIFAR 10 and risk for Adult should also be included in the main body of the paper.
- The experiment section only simulate the data prefiltering problem with label noise for classification setting. As motivated in the introduction, one important scenario is the next token prediction in the text setting. Note that in such case, each sample could contain some noisy portion (e.g. noise exists at a more granular level than sample level). Could the authors discuss how the proprosed method could be adapted to such cases?

---

> ### Author Response · Authors · 2026-03-13
> **Response to Reviewer asSb**
>
> We thank the reviewer for their careful and constructive comments.
>
> **Q1. Having a concrete algorithm describing the data prefiltering process, which is applied to the experiments will be helpful for audience to understand.**
>
> Thank you for the valuable suggestion! Please see “Further details on the prefiltering used in the experimental evaluation” in the shared response for a general pseudocode that we include in the updated manuscript.
>
> **Q2. The actual classification accuracy for CIFAR-10 and risk for Adult should also be included in the main body of the paper.**
>
> We again thank the reviewer for this suggestion! We have included this information in the updated version of the manuscript as Figure 1. Most notably, downstream CNNs yield between 52% and 62.5% in the presence of noise and no prefiltering, whereas our prefiltering procedure allows us to increase generalization accuracy to ~67% in the presence of prefiltering that is individually tailored to them. On the other hand, the learners in our experiment with tabular data show different improvements in F1 score in the presence of prefiltering, from little improvement (e.g., FFN shows no statistically significant improvement, remaining at 0.63) to significant improvement (e.g., SVM improves the average F1 score from 0.17 to 0.48). These results characterize the per-learner performance of the learner-specific prefiltering regimes, which serve as a reference point for computing the price of LARP. The price of LARP, quantifying the cost of jointly prefiltering for multiple downstream learners simultaneously, is reported in Figure 2, where we observe that it grows with contamination ratio and learner heterogeneity but remains small in absolute terms, supporting the practical viability of learner-agnostic prefiltering.
>
> **Q3. The experiment section only simulates the data prefiltering problem with label noise for classification setting.**
>
> We note that, beyond label noise, we also consider human annotation label noise (Appendix D.3), shortcut-based contamination simulating spurious correlations (Appendices D.2, D.8, D.9, D.10), and doubly-imbalanced data (Section 5.2), which creates unfairness in downstream models. Our empirical findings, in particular the price of LARP increasing with contamination ratio and learner heterogeneity, as well as its weak dependence on the dataset size, are consistent across the experiments.

---

> > ### Author Response · Authors · 2026-03-13
> > **Response to Reviewer asSb (Continued)**
> >
> > **Q4. As motivated in the introduction, one important scenario is the next token prediction in the text setting. Note that in such cases, each sample could contain some noisy portion (e.g., noise exists at a more granular level than sample level). Could the authors discuss how the proposed method could be adapted to such cases?**
> >
> > Thank you for the interesting point.
> >
> > First, we underline that our theoretical framework of the LARP problem is agnostic to how a “sample” is defined for the purposes of prefiltering. For text, we can define S as the set of all sentences, paragraphs, or documents in the corpus. We agree, however, that in tasks such as next-token prediction, the granularity of the noise (e.g., noisy tokens) might not match the scale of prefiltering (e.g., sample-level). This is similar to the shortcut experiments we present in Appendix D.2, where we incur spurious correlations in particular features of the training data according to the label, e.g., small patches in image data or modifying a particular integer feature. In such settings, we argue that it is not practical to remove this granular noise because this corrupts the context of the whole sample. For example, in our experiments with shortcuts, it does not make sense to delete the corner patch of the corrupted images, and the situation is similar for removing particular tokens from the middle of a document. Hence, there are three possible decisions a data curator can take for each sample: either (1) keep the sample as it is; (2) delete the entire sample; or (3) modify the sample. However, the latter is technically difficult to do at scale, since it requires applying a complex ML model (typically a generative one) on each training sample.  Additionally, modelling such modifications theoretically requires specifying the set of possible modifications and the rules for picking the optimal modification. Such modelling assumptions are often specific to the data modality and downstream use, making them impractical for web-scale datasets. Due to the additional technical complexity incurred, we believe that such generalization of LARP is worthy of future work. Under the assumption that data modification is not available, options (1) and (2) are fully covered by LARP.
> >
> > We note that common practices in web-scale datasets also restrict data preprocessing to the first two options above, discarding modifications. One notable example is the finemath-4+ dataset (Allal et al., 2025), created using the following procedure. Starting from the Common Crawl, the dataset curators finetune a classifier that gives each document a score from 1 to 5 according to “... its potential usefulness for studying mathematics up to high school and early undergraduate levels.” Then, the finemath-4+ dataset is created by retaining only documents that achieve a high score (4 or above).  This fits with our framework for removing samples, even though their utility can be measured more finely.
> >
> > **References:**
> >
> > Allal, Loubna Ben, et al. "SmolLM2: When Smol Goes Big--Data-Centric Training of a Small Language Model." arXiv preprint arXiv:2502.02737 (2025).
> >
> > Raffel, Colin, et al. "Exploring the limits of transfer learning with a unified text-to-text transformer." Journal of machine learning research 21.140 (2020): 1-67.

---

> ### Author Response · Authors · 2026-04-03
> **Follow-up on review discussion**
>
> Dear Reviewers,
>
> We are writing to kindly follow up on the status of our submission, as the deadline for final decisions (March 27) has passed. We remain available and happy to address any outstanding questions or concerns that may help move the discussion forward.
>
> Thank you for your continued effort in reviewing our work.
>
> Best regards,
> The Authors

---

### Review · Reviewer_HoeD · 2026-02-27

**Summary Of Contributions:**

The paper looks at characterizing risk behaviour under the setting of a data prefiltering mechanism using a common, possibly contamined dataset, applied to a fixed set of (heterogeneous) learners. The target is a prefiltering mechanism that minimizes the worst-case risk across learners, and the mean difference in per-learner tailored prefiltering vs. a common prefiltering is characterized as the prefiltering price to pay. Theoretical bounds characterizing behaviour of the prefiltering risk are given for particular chosen settings (set of Huber regression learners, under additional smoothness restriction; set of learners already satisfying individual upper risk bounds), and the tradeoff is empirically validated on CIFAR-10 and Adult datasets for small sets of learners under label contamination. An extension to slightly varying risk objectives is considered. Finally, an additional bound is given to motivate why learner-agnostic prefiltering might still be favourable over per-learner prefiltering by claiming higher utility.

**Additional Comments:**

- Is there a particular reason you start the setting by limiting the contamination rate to $\epsilon \in [0, 0.5)$ rather than $[0, 1]$?
- Instead of bounds on the learner-agnostic risk, could one also consider something like regret bounds on the price $P(F)$ of the filtering mechanism? Perhaps that would actually be more practical to quantify actual impact of prefiltering steps.
- What happens if the learner set is chosen to be really heterogeneous, does the framework then still make sense in practice? Clearly there is a strong sensitivity to this set, I am just wondering where the tradeoff of per-learner "cost of prefiltering" comes through. I mean, if we just fix a certain prefiltering strategy (e.g. quantile removal) then the only per-learner difference is in estimating the quantiles, how exactly does that induce any strong "cost" that we'd want to avoid? In App D7 the tested prefiltering strategy really doesn't seem to do much over no filtering, and overall performance is very consistent across both contamination types and learners (read: the contamination might be very benign and the learners are homogeneous), so all these theoretically highlighted tradeoffs are not entirely shining through in the experiments.

**Audience:**

Yes

**Audience Explanation:**

I am not very familiar with this area of research, but the results might be somewhat interesting to the data-centric machine learning community. However, the implications of this work for data prefiltering and its practicality seem fairly limited. The theoretical and practical results pertain to carefully and relatively narrowly selected learner sets (S4.1: set of Huber estimators with varying loss hyperparameter; S4.1.2: with additional smoothness condition; S4.2: learners satisfying already assumed upper bounds on individual risks and "oracle" prefiltering; Experiments: CNN with varying L2 regularization parameter; small batch of bagging and boosting algorithms; S5.2: identical model with objective with slightly varying tradeoff parameter), and aside of the "utility-driven" motivation (itself requiring additional assumptions) in S5.3 it overall remains unclear to me why its worth considering this kind of problem of "common prefiltering, fixed learners" from a practical standpoint.

**Broader Impact Concerns:**

No ethical concerns on my side, unless prefiltering is used specifically to target removal of particular dataset demographics in a negative way (e.g. removal of all data pertaining to a particular, undesirable political opinion or something like that)

**Claims And Evidence:**

Yes

**Claims Explanation:**

The paper is reasonably well-written to follow its arguments. The claims made are overall restricted to the particular design choices made (in particular, the theoretical bounds pertain exclusively to the handpicked settings) and don't seem overclaiming. The experimental results are fairly limited but, within the limitations of the setting, seem to align with some of the theoretical effects of parameter and design choices made. The paper is clear in repeatedly highlighting the cost of the considered common prefiltering framework.

**Requested Changes:**

- I think elaborating on connections to multi-distribution learning in S2 would also be suitable, in the sense that the idea of learner optimization w.r.t. worst-case risk across multiple fixed datasets could, I think, be reinterpreted as dataset optimization (prefiltering) w.r.t. worst-case risk across multiple fixed learners. At the very least the overall motivation and look of the objective is similar.
- I did not really understand how prefiltering is performed in practice in the experiments, as written in S5.1. Is it something like training a CNN to predict sample rankings, then removing the top $p$ fraction? Is it using ground-truth information following the oracle procedure (this would not make sense to me for testing)? What does it mean when you state "trained on the combined train and evaluation sets". I initially thought it would be akin to the quantile filtering, but I think that's not the case. This needs to be explained much better to connect the stated theory to the actual experiments run.
- The reason why I should still care about learner-agnostic prefiltering or when I should employ it in practice did not shine through to me. I understand the utility-driven motivation in S5.3, but it would be beneficial to have an experiment illustrating why such a prefiltering, despite individual learner risk increases, might be worth it to avoid per-learner "costly" prefiltering
- Similarly, it would be good to also consider different kinds of contaminations (e.g. high-leverage or contamination in the features) rather than just label noise. Is there a particular reason this was avoided and only focused on label noise?

---

> ### Author Response · Authors · 2026-03-13
> **Response to Reviewer HoeD**
>
> We thank the reviewer for their detailed review and valuable suggestions.
>
> **Q1. Motivation for considering the "common prefiltering, fixed learners" problem**
>
> Please see our shared response. Overall, we note that LARP is primarily a theoretical model for the task of data prefiltering from the point of view of a data provider. Additionally, in Section 5.3, we explore whether LARP can be beneficial from the point of view of multiple dataset users, splitting the costs of a single data prefiltering, compared to each of them filtering the dataset on their own.
>
> **Q2. Connections to multi-distribution learning**
>
> Please see our shared response.
>
> **Q3. I did not really understand how prefiltering is performed in practice in the experiments … Is it using ground-truth information following the oracle procedure (this would not make sense to me for testing)? What does it mean when you state "trained on the combined train and evaluation sets" …**
>
> Please see our shared response for the pseudocode of our general approach to data prefiltering in the experiments. In particular, all of our prefiltering procedures, except for the oracle one, do not assume ground-truth information. We do this to simulate a realistic prefiltering, done by a real-world data curator. Additionally, data splitting is done as follows. First, a small, clean risk evaluation dataset is separated for computing the final risk scores of downstream models. This data is assumed not to be available to the curator. Then, the remaining data is contaminated according to the empirical setup. Finally, the contaminated data passes through the prefiltering process as per Algorithm 1, and is then split into train and validation sets.
>
> **Q4. … it would be beneficial to have an experiment illustrating why such a prefiltering, despite individual learner risk increases, might be worth it to avoid per-learner "costly" prefiltering.**
>
> Our findings on the sublinear growth of the price of LARP in our empirical settings do indicate that the sufficient condition in Equation (9) for the learner-agnostic regime being preferable could be fulfilled for sufficiently large datasets (due to the weak dependence of the price of LARP on the dataset size).
>
> However, computing this condition in practice would amount to estimating the constant $C$, which abstracts the exchange rate between downstream accuracy loss and the computational cost of prefiltering. We expect the value of this constant to depend on the prefiltering algorithms and downstream applications, and so computing it would likely require thorough market research. For instance, the constant $C$ can be interpreted as the ratio of computational cost to the financial impact of a unit decrease in downstream model utility. As a simple example, we would expect that in medical settings, accuracy is important, which would bring the value $C$ down, increasing the threshold for the dataset size $n$. On the other hand, an application in which dataset curation is prohibitively expensive would imply a large value of $C$ and a relatively low threshold for $n$. Our game-theoretic setup aims to provide an asymptotic result, showing conditions under which learner-agnostic prefiltering is beneficial for sufficiently large datasets. This brings data prefiltering cost savings in addition to the overall benefit of providing principled data curation with guarantees for multiple downstream learners.
>
> **Q5. … it would be good to also consider different kinds of contaminations … rather than just label noise.**
>
> We do provide experiments with other types of data corruption in the original submission. In the supplementary material, we consider shortcuts (Appendix D.2, D.8, D.9, D.10), as well as human annotator label noise (Appendix D.3). Additionally, in Section 5.2, we study data imbalance and its impact on fairness and accuracy. Our empirical findings, in particular the price of LARP increasing with contamination ratio and learner heterogeneity, as well as its weak dependence on the dataset size, are consistent across the experiments.
>
> We also note that the theoretical framework in Section 4.1 studies LARP under the strong contamination model, as defined by Diakonikolas et al. (2023), which is different from the label noise used in experiments.

---

> > ### Author Response · Authors · 2026-03-13
> > **Response to Reviewer HoeD (Continued)**
> >
> > **Q6. Is there a particular reason you start the setting by limiting the contamination rate to $\epsilon \in [0,0.5)$ rather than $\epsilon \in [0,1]$?**
> >
> > Our rationale for choosing $\epsilon \in [0,0.5)$ stems from common limitations of robust learning. For example, one of the main takeaways of Proposition 1.7 in Diakonikolas et al. (2023) is that we need $\epsilon < 0.5$ to provide *any* meaningful guarantees on the learning procedure.
> >
> > That said, we agree that the condition is not necessarily needed for the broad definition of the setting. We clarify this in the revised manuscript, stating that while contamination can theoretically span $\epsilon \in [0,1]$, most contamination models would require $\epsilon < 0.5$ for meaningful robustness guarantees.
> >
> > **Q7. Instead of bounds on the learner-agnostic risk, could one also consider something like regret bounds on the price $P(F)$ of the filtering mechanism? Would that be more practical to quantify the actual impact of prefiltering steps?**
> >
> > We agree that giving upper bounds on $P(F)$ might provide applicable guarantees on the benefits of LARP. This motivates Corollary 5.2, in which we implicitly bound $P(F)$ using either of the assumptions to provide sufficient conditions for the benefit of LARP, without using the downstream accuracies (which would typically be unknown in advance).
> >
> > **Q8. What happens if the learner set is chosen to be really heterogeneous, does the framework then still make sense in practice?**
> >
> > First, we note that at least some assumptions on the learner set are needed. In particular, Lemma 4.1 indicates that otherwise we get $\Omega(1)$ learner-agnostic risk. This is also intuitive, as it is unreasonable to expect a data curator to accommodate ineffective learning algorithms.
> >
> > Given a reasonable learner set, heterogeneity indeed can lead to more pessimistic accuracy guarantees compared to guarantees for one specific learner. Nevertheless, these guarantees apply to all learners in the set, which might be a preferred property **by the data provider.**
> >
> > Additionally, the more pessimistic LARP guarantees may be offset by the reduced cost of data prefiltering **by the dataset users.** Specifically, in Section 5.3, we provide sufficient conditions for this to happen in a game-theoretic setting where dataset users can share data prefiltering costs under LARP. Since data prefiltering can be quite costly (see Q9), savings from shared prefiltering can bring higher utility compared to learner-specific filtering.
> >
> > **Q9. Where does the tradeoff of per-learner “cost of prefiltering” come through? If we just fix a certain prefiltering strategy (e.g., quantile removal) then the only per-learner difference is in estimating the quantiles, how exactly does that induce any strong “cost” that we'd want to avoid?**
> >
> > We explain why data prefiltering costs can be high even for a fixed prefiltering procedure.
> >
> > From the data provider's point of view, even under a fixed prefiltering procedure, the hyperparameters of that prefiltering still need to be set prior to public release. Referring back to our running examples in the Shared Response, in LAION-5B, which uses CLIP to check consistency between the initial pairs of image and caption, a datapoint is retained only if the cosine similarity between its caption and the CLIP-generated caption is above 0.28. Moving this threshold up or down can create different tradeoffs between data quality and quantity, and downstream learners might have different preferences. Thus, a single threshold will rarely be optimal for all learners simultaneously, which is what induces the price of LARP.
> >
> > On the other hand, if each downstream learner wanted to optimize this threshold for their specific model, they would need to (i) acquire the computational resources to run the scoring model (e.g., CLIP) over the entire dataset, and (ii) invest the expertise and experimentation effort to determine which threshold value best suits their particular use case. For large-scale datasets like Common Crawl, these computational and expert effort costs can be substantial, and they will be incurred independently by each learner in the learner-specific regime. It is precisely these repeated costs that learner-agnostic prefiltering avoids by selecting a single threshold and splitting the total cost across all downstream users.

---

> > > ### Author Response · Authors · 2026-03-13
> > > **Response to Reviewer HoeD (Continued)**
> > >
> > > **Q10. In App D7 the tested prefiltering strategy really doesn't seem to do much over no filtering, and overall performance is very consistent across both contamination types and learners (read: the contamination might be very benign, and the learners are homogeneous), so all these theoretically highlighted tradeoffs are not entirely shining through in the experiments.**
> > >
> > > We respectfully disagree that the prefiltering does not have an impact. In Figure 1, Adult dataset, comparing the no-prefiltering regime under noise (orange bars) with learner-specific prefiltering (green bars), we observe improvements across all seven learners, often recovering a large portion of the clean-data performance (blue bars). Similar patterns are visible in Figure 1, CIFAR-10 dataset, where learner-specific prefiltering (green curve) recovers roughly 5 percentage points of generalization accuracy compared to no prefiltering under noise (orange curve).
> > >
> > > Regarding the concern that learners appear homogeneous: the CIFAR-10 learner set varies the L2 regularization parameter over nearly an order of magnitude. We selected this range to ensure all learners are reasonable for the task (consistent with our theoretical requirements), while still capturing meaningful variation in noise sensitivity. Preliminary experiments showed that smaller values do not change training dynamics, whereas larger values cause the training to diverge. The Adult learner set spans seven qualitatively different ML algorithms (neural networks, SVMs, boosting, bagging, random forests). For both datasets, while these learners are all reasonable for the task (consistent with our theoretical requirements), they do exhibit different sensitivities to noise, as evidenced by the spread in performance visible in Figure 1. The price of LARP grows with the diversity of these sensitivities (Figure 2, left column), consistent with the theoretical prediction in Equation (2).
> > >
> > > **Q11. Scope of the experimental and theoretical analysis**
> > >
> > > We would like to take the opportunity to highlight the breadth of our experimental evaluation, much of which is presented in the supplementary material.
> > >
> > > **Data modalities and datasets** Our experiments span three distinct data modalities: scalar mean estimation (Appendix D.1), tabular data (Adult in Section 5.1; BAF in Section 5.2), and image data (CIFAR-10 in Section 5.1; CIFAR-10N in Appendix D.3; Tiny ImageNet in Appendix D.6). This covers a total of five real-world datasets in addition to the synthetic mean estimation setup.
> > >
> > > **Learner sets** We study learner sets that are heterogeneous in qualitatively different ways. For CIFAR-10, learners are CNNs differing in a continuous regularization parameter (L2 weight decay). For Adult, the learner set spans seven qualitatively different ML algorithms (neural networks, SVM, XGBoost, AdaBoost, LogitBoost, Bagging, Random Forest), representing a discrete and structurally diverse set. In the shortcuts setup (Appendices D.2, D.10), we also study learner sets parameterized by gradient reweighting coefficients and loss reweighting coefficients, respectively. For the fairness setup (Section 5.2), all learners use the same algorithm but differ in their risk functions, capturing heterogeneity in downstream objectives rather than in learning algorithms. Finally, for mean estimation (Section 4.1, Appendix D.1), learners are Huber estimators with varying robustness parameter $\delta$. Admittedly, some individual learner sets (e.g., CNNs with varying L2 regularization) are parameterized along a single axis, but this is a deliberate choice: it allows controlled study of how the price of LARP varies with learner heterogeneity, which is one of the main quantities of interest in our framework. Furthermore, Lemma 4.1 shows that the learner set must be restricted to reasonable learners, as otherwise the learner-agnostic risk admits an $\Omega(1)$ lower bound regardless of the prefiltering procedure.
> > >
> > > **Contamination models** Beyond uniform label noise, we study human annotation noise using the CIFAR-10N dataset (Appendix D.3), shortcut-based contamination simulating spurious correlations in both image and tabular data (Appendices D.2, D.8, D.9, D.10), double imbalance in the BAF dataset, which represents a subtle but prevalent form of data quality issue affecting fairness (Section 5.2), and the Huber contamination model for mean estimation (Section 4.1). We agree that studying additional contamination types, such as feature-space contamination or high-leverage points, is an interesting direction for future work.

---

> > > > ### Author Response · Authors · 2026-03-13
> > > > **Response to Reviewer HoeD (Continued)**
> > > >
> > > > **Q11. (Continued)**
> > > >
> > > > **Prefiltering procedures** We evaluate six distinct prefiltering procedures beyond the baseline: mean estimation prefiltering based on quantiles, z-score, and Stahel-Donoho outlyingness (Appendix D.1), loss-based prefiltering with expressive models (Section 5.1), Confident Learning (Northcutt et al., 2021; see Appendix D.4), oracle prefiltering (Appendix D.5), and time-to-low-loss filtering (Appendix D.9), as well as fairness-oriented reweighting (Section 5.2). Our results show consistent trends across all procedures, supporting the generality of our findings.
> > > >
> > > > Our theoretical analysis indeed covers specific settings only (Huber estimators for mean estimation in Section 4.1, and the PAC-style oracle setup in Section 4.2). However, as noted in the Shared Response, we see this as a necessary first step in analyzing the LARP problem. In particular, mean estimation is a standard setting of interest in robust statistics and ML literature, and the Huber estimators are a canonical example of estimators with varying degrees of robustness. Extending theoretical guarantees to broader settings such as regression and classification is an exciting direction for future work, as discussed in Section 6.
> > > >
> > > > **References:**
> > > >
> > > > Diakonikolas, Ilias, and Daniel M. Kane. Algorithmic high-dimensional robust statistics. Cambridge University Press, 2023.
> > > >
> > > > Northcutt, Curtis, Lu Jiang, and Isaac Chuang. "Confident learning: Estimating uncertainty in dataset labels." Journal of Artificial Intelligence Research 70 (2021): 1373-1411.

---

> ### Author Response · Authors · 2026-04-03
> **Follow-up on review discussion**
>
> Dear Reviewers,
>
> We are writing to kindly follow up on the status of our submission, as the deadline for final decisions (March 27) has passed. We remain available and happy to address any outstanding questions or concerns that may help move the discussion forward.
>
> Thank you for your continued effort in reviewing our work.
>
> Best regards,
> The Authors

---

### Author Response · Authors · 2026-03-13
**Shared Response**

We thank all reviewers for their constructive feedback and engaging questions.

Based on the received comments, we identified several common points that we address below. We have updated our manuscript to reflect these changes. We also respond to the remaining points with individual official comments under each review. We welcome any further questions and are happy to provide additional clarifications during the remainder of the discussion period.

**Purpose and scope of our work**

To address concerns from Reviewers HoeD (regarding the usefulness of the LARP framework) and wYKL (regarding what LARP is), we further clarify the purpose of our work.

In our work, we formalize a data prefiltering objective for a data provider that seeks to release a public dataset for ML models training. Since such datasets may be adopted by various downstream practitioners, it is desirable that such prefiltering ensures performance guarantees for a broad specification of learners that may be applied on top. The task of LARP, i.e., minimizing a worst-case loss over a set of learners, is one natural formalization of this objective that we propose. We see our main contribution in proposing the mathematical framework of LARP and analyzing it in several natural theoretical and experimental settings, rather than providing some end-to-end solution to the problem of public data prefiltering.

**Motivation for studying prefiltering of public data**

Our study is motivated by the fact that public datasets released specifically for ML training are now ubiquitous. Prefiltering such data is thus of central importance for facilitating downstream ML training. For example, the LAION-5B dataset (Schuhmann et al., 2022) has been found to contain synthetic images of poor quality, potentially harming downstream learning (Alemohammad et al., 2023). Similar data quality issues are also found in crowdsourced datasets (Yan et al., 2014; Yu et al., 2018). On the other hand, overzealous prefiltering can also have negative effects, either due to insufficient remaining data or unfair representation of minorities. For example, a filter applied to the C4 dataset (Raffel et al., 2020) was reported to have reduced the amount of texts in dialects and hence overall fairness (Dodge et al., 2021). Finally, the issue of collecting and preprocessing datasets is of central importance in AI for science (Aït-Sahalia et al., 2010; Pechenizkiy et al., 2006; Liu et al., 2020).

We note that prefiltering public data to facilitate robust model training is not only intuitively desirable but also aligns with recent calls for transparency in dataset creation (Gebru et al., 2021) and for data-centric strategies in developing robust and ethical AI systems (Liang et al., 2022). Additionally, data prefiltering at the scale of SOTA ML benchmarks and training datasets can be computationally expensive, due to their size, and also technically challenging, due to various types of data issues that can arise in large datasets. Therefore, in many settings, data providers may be more capable of performing data prefiltering than individual dataset users.

---

> ### Author Response · Authors · 2026-03-13
> **Shared Response (Continued)**
>
> **Benefits of LARP**
>
> Both reviewers HoeD and wYKL inquired about when LARP can be useful, compared to standard existing methods for data filtering.
>
> First, we note that LARP is one way of formalizing the problem of principled data prefiltering by the data provider, with the goal of making the dataset more suitable for downstream ML training. This is conceptually different from the common robust ML setting, where prefiltering is one possible step that each learner (e.g., a company that trains a model based on such data) can use to improve its resulting model. In many settings, prefiltering being done by the data provider can be preferable, both because it can improve the quality of the provided dataset and because some end users may not have the computational resources and expert know-how to perform prefiltering to the same extent. Overall, our theoretical and empirical results indicate that it is possible to provide meaningful LARP guarantees and improvements in empirical performance across several natural families of learners.
>
> Additionally, in Section 5.3, we analyze the potential benefits of learner-agnostic prefiltering in a setting where the **entities that will use the dataset** split the cost of the (single) prefiltering under LARP. Within the game-theoretic model in that section, we derive sufficient conditions under which LARP is beneficial, also from the perspective of saving cost from repeated data curation (that would occur in learner-specific filtering). The cost difference between the two regimes studied there comes from the split of a single prefiltering cost across all learners, as opposed to each learner independently bearing the full cost of prefiltering. Still, we want to emphasize that LARP can be independently beneficial, as individual consumers of the dataset can find prefiltering too computationally complex and technically difficult to do in the first place.
>
> **Relationship to DRO and multi-distribution learning**
>
> We thank Reviewers HoeD and wYKL for highlighting the conceptual similarities between LARP and established robust frameworks, specifically multi-distribution learning (MDL) (Blum et al., 2017; Haghtalab et al., 2022) and distributionally-robust optimization (DRO) (Ben-Tal et al., 2013; Sagawa et al., 2020). Both reviewers are correct to observe that the minimax structure of our objective, namely optimizing for the worst-case risk over a set, conceptually matches these paradigms.
>
> We provide a comparison to these frameworks below and have included it in the updated version of our manuscript.
>
> At a high level, the fundamental difference between LARP and these frameworks lies in what is being optimized. Frameworks like DRO and MDL optimize a single learner to ensure robustness across multiple data distributions (for MDL) or perturbed test sets (for DRO). In LARP, on the other hand, one optimizes over what data to keep (through the prefiltering mechanism) to ensure robustness across multiple downstream learners.
>
> **MDL** In the multi-distribution learning framework of Blum et al. (2017), the theoretical model assumes multiple input distributions. LARP operates in an inverse setting: we assume a single contaminated input, and the objective is defined as a worst-case risk over the fixed learner set. On the technical side, reducing LARP to that framework would require a mathematical toolbox that translates learner set heterogeneity into heterogeneity in the data distribution. We believe developing such a mathematical toolbox to be challenging for many reasons, most notably because such techniques could be sensitive to the learner sets as functions, as well as to the prefiltering procedure. Nevertheless, we agree that such an analysis could be useful for deriving guarantees for LARP and have added it as a natural step for future work.
>
>
> **DRO** Because LARP deals with a single input distribution and is concerned with the minimax bound over a set of downstream risks, we agree that the problem bears a strong resemblance to DRO. In DRO, one can study the tradeoff between the accuracy of a model and the validity of its result in the presence of small distribution shifts. In LARP, we study the tradeoff between the accuracy of individual downstream learners and the validity of our guarantees across the entire learner set.  Just as the robust optimization literature considers the “price of robustness” (Bertsimas et al., 2004), quantifying the reduction in a model's standard accuracy due to increased distributional robustness, LARP incurs a “price of LARP” due to the prefiltering being protective for a larger variety of learners. Furthermore, in the context of Section 5.3, our goal in accepting this price is the split of the curation cost, as obtaining a single dataset that is relatively good for many models may be economically preferable to every practitioner cleaning the data themselves.

---

> > ### Author Response · Authors · 2026-03-13
> > **Shared Response (Continued)**
> >
> > **Further details on the prefiltering used in the experimental evaluation**
> >
> > ---
> > > **Algorithm 1: General Prefiltering Procedure**
> > ---
> > > **Require:** Provider Dataset $S$, fraction $p \in [0, 1)$, scoring function $g(\cdot, \cdot)$, and scoring model/estimator $\mathcal{M}$
> > > $\mathcal{M}\_\theta \leftarrow \text{Fit}(S, \mathcal{M})$  $\qquad \triangleright$ *Fit the scoring model*
> > > $\textit{Scores} \leftarrow \emptyset$
> > > **for** each $z_i \in S$ **do**
> > > $\quad$ $s_i \leftarrow g(z_i, \mathcal{M}_\theta)$  $\qquad \triangleright$ *Compute outlyingness (e.g., loss or z-score)*
> > > $\quad$ $\textit{Scores}.\text{append}(s_i)$
> > > **end for**
> > > $\tau \leftarrow \text{Quantile}(\textit{Scores},\ 1-p)$ $\qquad \triangleright$ *Find the cutoff threshold*
> > > $S' \leftarrow \\{ z\_i \in S \mid g(z\_i, \mathcal{M}\_\theta) < \tau \\} $   $\qquad \triangleright$ *Keep only points below threshold*
> > > $S'\_{\text{train}},\ S'\_{\text{val}} \leftarrow \text{TrainValSplit}(S')$   $\qquad \triangleright$ *Split into train and validation sets*
> > > **return** $S'\_{\text{train}},\ S'\_{\text{val}}$
> > ---
> >
> > To clarify the prefiltering procedures used in our experiments, we present Algorithm 1. This procedure represents the general structure of the prefiltering procedures used in our experiments.
> >
> > We clarify that what we previously called the test dataset is a clean dataset that we use to calculate the price of LARP. This data will typically not be available to a model prefiltering procedure, but it is crucial for validating our findings because our risk is a generalization risk. To reduce the confusion stemming from this naming, we have adjusted the paper to call it **risk evaluation data**. The rest of the dataset is considered to be the unfiltered data available to the dataset provider/collector. We simulate contamination on this dataset to simulate noisy real-world data that the provider typically obtains during data collection. This dataset is first run through Algorithm 1 to be prefiltered, and then split into train and validation for each learner to use.
> >
> > Given our dataset $S$, the algorithm first fits a scoring model $\mathcal{M}$, e.g., in image data experiments in Section 5.1, we train ResNet-9 for a fixed number of iterations on $S$. Then Algorithm 1 executes a scoring function $g$ on this model. For example, for the experiments in Section 5.1, we choose $g(z, \mathcal{M}\_\theta)$ as the loss of $\mathcal{M}\_\theta$ on the datapoint $z$. This is motivated by the fact that neural networks tend to learn useful features before overfitting to label noise (Zhang et al., 2017; Arpit et al., 2017), hence noisy points would tend to have higher losses during early stages of training. The algorithm then gathers all the scores, computes a threshold $\tau$ as the $(1-p)$-th percentile of all scores, and filters out all points above that quantile threshold. Finally, the prefiltered dataset $S’$ is split into train and validation sets, the latter being utilized for early stopping.
> >
> > This description matches the general structure of the prefiltering procedures used for the experiments with label noise (Section 5.1, Appendices D.4, D.6), human label noise (Appendix D.3), and shortcuts (Appendices D.2, D.8, D.9, D.10). For example, in Section 5.1, we have the following realizations:
> >  * In label noise experiments on CIFAR-10, $\mathcal{M}$ is ResNet-9, and  $g(z, \mathcal{M}\_\theta)$ is the loss of $\mathcal{M}\_\theta$ on $z$.
> >  * In label noise experiments on Adult, $\mathcal{M}$ is 2-layer NN, and  $g(z, \mathcal{M}\_\theta)$ is the loss of $\mathcal{M}\_\theta$ on $z$.
> >  * In shortcut experiments on CIFAR-10, $\mathcal{M}$ is our custom CNN, and  $g(z, \mathcal{M}\_\theta)$ is the negative of the loss of $\mathcal{M}\_\theta$ on $z$. We take the negative because we expect shortcut-contaminated data to have lower losses during training.
> >  * In shortcut experiments on Adult, $\mathcal{M}$ is Random Forest, and  $g(z, \mathcal{M}\_\theta)$ is the negative of the loss of $\mathcal{M}\_\theta$ on $z$.
> >
> > For further dataset-specific details, please see the corresponding appendices. This algorithm also covers the prefiltering procedure presented in Section 5.2, with some modifications. We refer the interested reader to Appendix E in our manuscript for additional discussion.

---

> > > ### Author Response · Authors · 2026-03-13
> > > **Shared Response (Continued)**
> > >
> > > **Further details on the prefiltering used in the experimental evaluation (Continued)**
> > >
> > > In addition, Algorithm 1 captures the high-level structure of the prefiltering procedures used for mean estimation (Section 4.1, Appendix D.1). It also covers the oracle prefiltering procedures (Section 4.2, Appendix D.5) under the idealistic assumption that the prefiltering procedure, through the scoring function $g$, also has access to whether each data point is contaminated or not.
> > >
> > > Finally, Algorithm 1 bears resemblance to standard data filtering pipelines used to curate modern large-scale datasets. For example, LAION-5B (Schuhmann et al., 2022) uses a pre-trained CLIP model ($\mathcal{M}$) (Radford et al., 2021) to filter image-text pairs. Similarly, CCNet (Wenzek et al., 2020) employs a language identification model ($\mathcal{M}$) to curate the Colossal Clean Crawled Corpus. Therefore, varying the definitions of $\mathcal{M}$ and $g$ in Algorithm 1 can describe common heuristic prefiltering practices of foundation-model datasets, similarly to the loss-based filtering used in our experimental analysis. Note that while real-world use cases set constant thresholds on the scores produced by $g$, this is equivalent to setting the fraction $p$ of datapoints that should be filtered out, as in Algorithm 1.
> > >
> > > In order to improve the clarity of our experiments, we provide an updated version of the manuscript containing Algorithm 1 as well as the additional description and running examples.
> > >
> > >
> > > **References:**
> > >
> > > Alemohammad, Sina, et al. "Self-consuming generative models go MAD." The Twelfth International Conference on Learning Representations. 2023.
> > >
> > > Arpit, Devansh, et al. "A closer look at memorization in deep networks." International conference on machine learning. PMLR, 2017.
> > >
> > > Aït-Sahalia, Yacine, Jianqing Fan, and Dacheng Xiu. "High-frequency covariance estimates with noisy and asynchronous financial data." Journal of the American Statistical Association 105.492 (2010): 1504-1517.
> > >
> > > Ben-Tal, Aharon, et al. "Robust solutions of optimization problems affected by uncertain probabilities." Management Science 59.2 (2013): 341-357.
> > >
> > >
> > > Bertsimas, Dimitris, and Melvyn Sim. "The price of robustness." Operations Research 52.1 (2004): 35-53.
> > >
> > > Blum, Avrim, et al. "Collaborative PAC learning." Advances in Neural Information Processing Systems 30 (2017).
> > >
> > > Dodge, Jesse, et al. "Documenting Large Webtext Corpora: A Case Study on the Colossal Clean Crawled Corpus." Proceedings of the 2021 Conference on Empirical Methods in Natural Language Processing. 2021.
> > >
> > > Gebru, Timnit, et al. "Datasheets for datasets." Communications of the ACM 64.12 (2021): 86-92.
> > >
> > > Haghtalab, Nika, Michael Jordan, and Eric Zhao. "On-demand sampling: Learning optimally from multiple distributions." Advances in Neural Information Processing Systems 35 (2022): 406-419.
> > >
> > > Liang, Weixin, et al. "Advances, challenges and opportunities in creating data for trustworthy AI." Nature Machine Intelligence 4.8 (2022): 669-677.
> > >
> > > Liu, Sheng, et al. "On the design of convolutional neural networks for automatic detection of Alzheimer’s disease." Machine learning for health workshop. PMLR, 2020.
> > >
> > > Pechenizkiy, Mykola, et al. "Class noise and supervised learning in medical domains: The effect of feature extraction." 19th IEEE symposium on computer-based medical systems (CBMS'06). IEEE, 2006.
> > >
> > > Radford, Alec, et al. "Learning transferable visual models from natural language supervision." International conference on machine learning. PMLR, 2021.
> > >
> > > Raffel, Colin, et al. "Exploring the limits of transfer learning with a unified text-to-text transformer." Journal of machine learning research 21.140 (2020): 1-67.
> > >
> > > Sagawa, Shiori, et al. "Distributionally robust neural networks for group shifts: On the importance of regularization for worst-case generalization." International Conference on Learning Representations. 2020.
> > >
> > >
> > > Schuhmann, Christoph, et al. "LAION-5B: An open large-scale dataset for training next generation image-text models." Advances in neural information processing systems 35 (2022): 25278-25294.
> > >
> > > Wenzek, Guillaume, et al. "CCNet: Extracting high quality monolingual datasets from web crawl data." Proceedings of the twelfth language resources and evaluation conference. 2020.
> > >
> > > Yalcin, Ata, et al. "Fair for a few: Improving Fairness in Doubly Imbalanced Datasets." arXiv preprint arXiv:2506.14306 (2025).
> > >
> > > Yan, Yan, et al. "Learning from multiple annotators with varying expertise." Machine learning 95.3 (2014): 291-327.
> > >
> > > Yu, Xiyu, et al. "Learning with biased complementary labels." Proceedings of the European conference on computer vision (ECCV). 2018.
> > >
> > > Zhang, Chiyuan, et al. "Understanding deep learning requires rethinking generalization." International Conference on Learning Representations. 2017.

---

### Decision · Action_Editor_f1D9 · 2026-05-04

**Recommendation:** Accept as is

**Audience:**

Yes

**Audience Explanation:**

Yes, the findings of this paper are of significant interest to the data-centric sub-community of machine learning. In the ultra-large-scale data regime, filtering needs to be automatic and such studies area timely. The shift of robustness from model-side to data-side is also timely as these two are typically handled by seperate teams, and some-times separate entities.

**Claims And Evidence:**

Yes

**Claims Explanation:**

The submission provides a set of theoretical and empirical evidence to support its claims. The mathematical framework is rigorous, offering specific theoretical bounds for various settings under some reasonable restrictions. Although the setting is interesting (scalar mean estimation and Huber estimators), it is still somewhat limited. The reviewers generally agreed that these results are technically sound and that the authors have also been careful not to overclaim, explicitly acknowledging the limitations of their chosen settings.

In the empirical evaluation, the authors validated their findings using standard datasets like CIFAR-10 and Adult under label contamination. While some reviewers initially noted that the experimental scope was relatively narrow, the authors’ revisions clarified the implementation details and demonstrated that the observed effects align with their theoretical predictions. The empirical study clearly suffices to verify the proposed theoretical study.

The manuscript is reviewed by expert reviewers and they raised some significant issues. After some discussions and a revision by the authors, manuscript received unanimous accept decision from the reviewers. I support this decision as well.

---

> ### Author Response · Authors · 2026-06-02
> **Camera Ready Revision**
>
> Dear Editors-in-Chief, Action Editor, and Reviewers,
>
> We thank you for the valuable feedback and the positive recommendation. We have uploaded the camera ready version of our submission. Our main changes consist of polishing the text and updating references. We have also added links to the video presentation and the GitHub repository containing the code.
>
> Best regards,
> The Authors